# Genetic and functional characterization of AMH Signaling in Zebrafish - Evidence for Roles of Amh-Bmpr2a-Bmpr1bb Pathway in Controlling Gonadal Homeostasis

Yiming Yue[1], Chu Zeng[1], Xin Zhang ![ORCID][1], Chao Bian[2], Zhiwei Zhang[1], Kun Wu[1,3], Weiting Chen[1,4], Xianqing Zhou[5], Ling Lu[6], Nana Ai[1], Wei Ge ![ORCID][1]*

1 Department of Biomedical Sciences and Centre of Reproduction, Development and Aging (CRDA), Faculty of Health Sciences, University of Macau, Macau, China, 2 Laboratory of Aquatic Genomics, College of Life Sciences and Oceanography, Shenzhen University, Shenzhen, China, 3 Southern Marine Science and Engineering Guangdong Laboratory (Zhuhai), State Key Laboratory for Biocontrol, Sun Yat-sen University, Zhuhai, China, 4 Guangdong Provincial Key Laboratory of Conservation and Precision Utilization of Characteristic Agricultural Resources in Mountainous Area, School of Life Sciences, Jiaying University, Meizhou, China, 5 Department of Toxicology and Hygienic Chemistry, School of Public Health, Capital Medical University, Beijing, China, 6 Key Laboratory of Marine Drugs, The Ministry of Education of China, School of Medicine and Pharmacy, Ocean University of China, Qingdao, China

* weige@um.edu.mo, gezebrafish@gmail.com

## Abstract

Anti-Müllerian hormone (AMH), a member of the TGF-β superfamily, plays a crucial role in Müllerian duct regression in mammals. AMH signals through a specific type II receptor (AMHRII) and a type I receptor to activate the downstream Smad1/5/9 signaling pathway. Interestingly, non-mammalian vertebrates, including teleosts, also express AMH despite lacking Müllerian ducts. Accumulating evidence indicates that AMH influences gonadal development and function across vertebrates. Intriguingly, zebrafish, a popular model organism, possesses AMH (Amh/amh) but lacks specific type II receptor (Amhr2). Based on our previous studies and data from others, we propose that Amh may signal through a novel pathway in zebrafish involving the BMP type II receptor Bmpr2a and type I receptor Bmpr1bb. In this study, we provide genetic and functional evidence for the existence of the Amh-Bmpr2a-Bmpr1bb signaling pathway in zebrafish and its role in controlling gonadal homeostasis. Our experimental data excluded the participation of Bmpr2b and Bmpr1ba, paralogues of Bmpr2a and Bmpr1bb respectively, in Amh signaling. Additionally, we also provide genetic evidence that the phenotypes exhibited by *amh, bmpr2a*, and *bmpr1bb* mutants, *e.g.*, gonadal hypertrophy, were all dependent on gonadotropin signaling, and that the two gonadotropins (FSH and LH) could both drive the development of hypertrophic gonadal growth resulting from dysfunction in the Amh-Bmpr2a-Bmpr1bb signaling pathway, with FSH showing a more immediate effect in females. In summary, the present study provides comprehensive evidence for Amh signaling through

**Data availability statement:** All relevant data are within the manuscript and its Supporting Information files.

**Funding:** This research was supported by funding from the University of Macau (MYRG-GRG2023-00144-FHS-UMDF, MYRG-GRG2024-00191-FHS, MYRG-GRG2025-00220-FHS, CPG2024-00030-FHS, and CPG2025-00037-FHS) and The Macau Fund for Development of Science and Technology (FDCT0086/2022/AFJ and FDCT-NSFC Joint Project 0086/2022/AFJ) to WG. The funders had no role in study design, data collection and analysis, decision to publish, or preparation of the manuscript.

**Competing interests:** The authors have declared that no competing interests exist.

Bmpr2a-Bmpr1bb pathway and its interplay with gonadotropins in controlling germ cell proliferation and differentiation, thereby maintaining gonadal homeostasis.

## Author summary

Anti-Müllerian hormone (AMH) is a well-known protein essential for male sexual development in humans and other mammals. However, its function in lower vertebrates like fish, which lack the specific structures AMH regulates in mammals, has remained a puzzle. Furthermore, zebrafish possess the AMH protein (Amh) but surprisingly lack the specific receptor found in other species, which raises the question of how this hormone functions. In this study, we solved this mystery by identifying a novel signaling pathway that mediates Amh actions in zebrafish. We demonstrated that zebrafish Amh utilizes two alternative receptors, Bmpr2a and Bmpr1bb, to transmit its signals. By using gene editing technology to disrupt these receptors, we observed that the mutant fish without Bmpr2a and Bmpr1bb developed excessively enlarged gonads, mirroring the effects of losing Amh itself. We further demonstrated that this pathway acts as a critical "brake" on gonadotropins, the master hormones that drive reproduction. Without this brake, the gonads are overstimulated and grow uncontrollably. Our findings not only clarify the unique way Amh functions in zebrafish but also highlight a sophisticated evolutionary mechanism where an ancient hormone signals through different receptors to maintain reproductive balance and gonadal health.

## Introduction

Gonadal development and function are crucial to vertebrate reproduction, involving complex interplays between endocrine hormones and paracrine factors. The transforming growth factor β (TGF-β) superfamily, known for its molecular and functional diversity in various developmental and physiological processes, plays significant roles in reproductive development and function [1,2]. Despite its ligand diversity, the TGF-β superfamily shares conserved signaling pathways involving type II and type I receptors, both being single cross-membrane serine/threonine kinase receptors. In mammals, five type II receptors have been identified for TGF-β superfamily signaling, including TGFBR2 for TGF-βs, ACVR2A and ACVR2B for activins, BMPR2 for bone morphogenetic proteins (BMPs), and AMHR2 for anti-Müllerian hormone (AMH) [3,4]. Additionally, seven type I receptors, known as activin receptor-like kinases (ALK1–7), have been identified [5]. The ligands of TGF-β superfamily act by binding to a specific type II receptor first followed by recruiting a less specific type I receptor, which first phosphorylates intracellular receptor-regulated Smad proteins (R-Smads: Smad1, 2, 3, 5 and 8/9), then forming heteromeric complexes with a common Smad (Co-Smad: Smad4). The Smad complexes subsequently migrate to the nucleus to regulate expression of target genes, thus inducing various biological responses [6,7]. The

TGF-β superfamily comprises more than 30 ligand genes [8]; among them, AMH has been a focus of research due to its multifaceted roles in reproduction of both males and females.

AMH, also known as Müllerian-inhibiting substance (MIS), is well known for its classical role of suppressing the development of Müllerian ducts in male embryos [9,10]. Mutations in AMH and its type II receptor AMHR2 result in Persistent Müllerian Duct Syndrome (PMDS) in both humans and murine models [11]. In adults, AMH is secreted by Sertoli cells in the testis and granulosa cells in the ovary, and it plays crucial roles in gametogenesis of both sexes [12]. In females, AMH is primarily produced by the ovarian follicles at the primary stage onwards but not primordial follicles, and it influences the depletion rate of the ovarian reserve [13], therefore serving as a marker for assessing ovarian reserve, forecasting reproductive lifespan, and diagnosing disorders such as polycystic ovary syndrome (PCOS) [14–16]. AMH exerts significant effects throughout folliculogenesis, particularly inhibiting the initial recruitment of primordial follicles so as to influence the ovarian reserve [13]. Mutation of AMH in mice resulted in increased number of preantral and small antral follicles compared to the wild-type control [17,18]. It has been reported that part of mechanism by which AMH regulates folliculogenesis is to diminish the sensitivity of growing follicles to gonadotropins (follicle-stimulating hormone, FSH; luteinizing hormone, LH), especially FSH, which affects follicle selection [19]. In males, AMH has been implicated in the process of testicular descent in fetal development and its onset of expression correlates well with the differentiation of the Sertoli cells. In adults, AMH continues to be expressed, and its serum level is used as a marker for testis function [20]. Recent research in mice has further revealed the role of Sertoli cell-derived AMH in sustaining Sertoli cell identity in testis [21].

Like other TGF-β family members, AMH acts by interacting with its specific type II receptor, AMHR2, followed by recruiting a type I receptor, leading to the activation of Smad1/5/8 (now Smad1/5/9) [22]. Several type I receptors have been suggested to participate in AMH signaling with AMHR2, including ACVR1, BMPR1A, and BMPR1B [23–25].

Although teleost fish lack Müllerian ducts, AMH orthologs (Amh/amh) have been documented in more than 20 species [26]; however, the Amh type II receptor Amhr2 has only been reported in a few species, including medaka, Nile tilapia, black porgy, coelacanth and tiger pufferfish [26]. Interestingly, an extra copy of amh gene (amhy) has been localized on the Y chromosome in several species: Patagonian pejerrey, Nile tilapia, Northern pike, Japanese flounder, and black rockfish, where it serves as a sex-determining gene [26–29]. In medaka, mutation of Amh receptor amhr2 resulted in sterility with excessive germ cell proliferation in both sexes, leading to severe gonadal hypertrophy [30]. In tilapia, female heterozygous mutations in amh or amhr2 increased primary growth follicles and reduced fertility, whereas homozygous mutations caused ovarian hypertrophy and disrupted follicular transition to vitellogenic growth in the ovary [31]. Males with amh mutations exhibited increased spermatogonial proliferation, which could be compensated for by the presence of amhy [32].

Amh has also been well studied in zebrafish where it is primarily expressed in the Sertoli cells in the testis and granulosa cells in the ovary [33]. Similar to the phenotypes reported in medaka and tilapia, the amh mutant of zebrafish also displayed severe gonadal hypertrophy with increased germ cell proliferation but decreased differentiation, indicating its crucial role in gonadal homeostasis [34–36]. These phenotypes agree well with an early in vitro study showing that Amh suppressed spermatogonial proliferation in cultured zebrafish testis [37]. Interestingly, unlike medaka and tilapia, zebrafish lacks an Amhr2 homologue in its genome, raising an intriguing question about how Amh signals in this species. Phylogenetic analysis revealed existence of two variants of BMP type II receptors (BMPRII) in zebrafish, Bmpr2a and Bmpr2b, both closely related to Amhr2. Although they both respond to BMPs [38], our recent genetic analysis strongly suggests that Bmpr2a, but not Bmpr2b, acts as a type II receptor for Amh signaling in zebrafish, as mutant of bmpr2a, but not bmpr2b, phenocopied the amh mutant [38]. Regarding the type I receptor(s) involved in Amh signaling in zebrafish gonads, the information is limited. An earlier study reported that a point mutation in bmpr1bb gene (one of the two BMPR1B gene paralogues in zebrafish) also resulted in gonadal hypertrophy with accumulation of early germ cells [39]. This phenocopied both amh and bmpr2a mutants, suggesting a potential role for Bmpr1bb in Amh signaling. Although these genetic studies point to the existence of Amh-Bmpr2a-Bmpr1bb signaling pathway in zebrafish, further evidence is

needed to verify its functionality. Moreover, the potential roles of Bmpr2b and Bmpr1ba, which are paralogues of Bmpr2a and Bmpr1bb respectively, in Amh signaling are also interesting to explore.

As proposed for AMH in mammals [19,40], our genetic analysis in zebrafish also suggests a powerful inhibitory effect of Amh on gonadotropin signaling in both ovary and testis, effectively keeping gonadotropin actions in check [38]. Simultaneous mutation of gonadotropin receptors (*fshr* and *lhcgr*) could completely abolish the phenotype of gonadal hypertrophy induced by *amh* mutation [38]. However, it remains unknown whether the gonadal hypertrophy resulting from *bmpr2a* and *bmpr1bb* mutations is also dependent on gonadotropin signaling, and whether the two gonadotropins, FSH and LH, play differential roles in the process.

To address these issues, we conducted this study with aims to validate the proposed Amh-Bmpr2a-Bmpr1bb pathway in controlling gonadal homeostasis and elucidate the intricate interactions between Amh and gonadotropin signaling in the gonads. Our findings provide comprehensive evidence for the significance of Amh function in gonadal homeostasis and its signaling mechanism in zebrafish.

## Results

### Spatiotemporal expression profiles of *amh* and its potential receptors (*bmpr2a/2b* and *bmpr1ba/bmpr1bb*) in follicles

Understanding the spatial distribution of Amh and its potential receptors within the follicle, especially between germ cells (oocyte) and somatic cells in the follicle layer (granulosa and theca cells), is essential to elucidate their modes of action and interaction. To achieve this, we mechanically separated the follicle layers and oocytes of FG follicles from sexually mature females according to a previously established protocol [41]. We then assessed the expression of *amh* and its potential receptors, type II (*bmpr2a* and *bmpr2b*) and type I (*bmpr1ba* and *bmpr1bb*), in both the isolated follicle layers and oocytes using semiquantitative RT-PCR. Luteinizing hormone receptor (*lhcgr*) and growth differentiation factor 9 (*gdf9*) were used as molecular markers for follicle layers and oocytes, respectively.

Consistent with previous in situ hybridization studies [33,35,42], we found that *amh* was exclusively expressed in the follicle layer. Similarly, the type II receptors *bmpr2a* and *bmpr2b* were also expressed in the follicle layer, confirming our previous findings [43,44]. Regarding the type I receptors, *bmpr1bb* expression was confined to the follicle layer together with *bmpr2a* and *bmpr2b*. In contrast, *bmpr1ba* expression could be detected in both compartments, with a higher expression in the oocyte (Fig 1A).

We next analyzed the temporal expression patterns of *amh, bmpr2a, bmpr2b, bmpr1ba* and *bmpr1bb* during folliculogenesis using RT-qPCR, with FSH receptor (*fshr*) and LH receptor (*lhcgr*) serving as references [45]. The *lhcgr* expression increased progressively during vitellogenic growth from EV stage, reaching maximum levels at FG stage. The expression of *fshr,* however, rose from PV stage, peaked at EV-MV stages, and then decreased at LV and FG stages before maturation (Fig 1B). In comparison, the expression of *amh* was low at PG stage but surged to a peak level at PV stage, coinciding with the PG-PV transition, which marks the shift from the slow-growing primary growth (PG) phase to fast-growing secondary growth (SG) phase. This peak expression at the PV stage was followed by a progressive and rapid decline during vitellogenic growth (EV-LV), reaching its lowest level at FG stage. This expression profile is somewhat similar to that observed in mammals [46]. The expression of the type II receptors, *bmpr2a* and *bmpr2b*, increased steadily and significantly from the PG stage onwards, both reaching the peak levels at FG stage, consistent with our previous report [47]. In comparison, the type I receptors *bmpr1ba* and *bmpr1bb* exhibited relatively constant expression throughout follicle development (Fig 1C).

### Expression and co-expression of *amh, bmpr2a/b* and *bmpr1ba/b* in somatic follicle cells

Although our results indicated that *amh* and its potential receptors were expressed in the follicle layer, their precise localization in different follicle cell populations remained unclear. To address this, we used a recently published single-cell

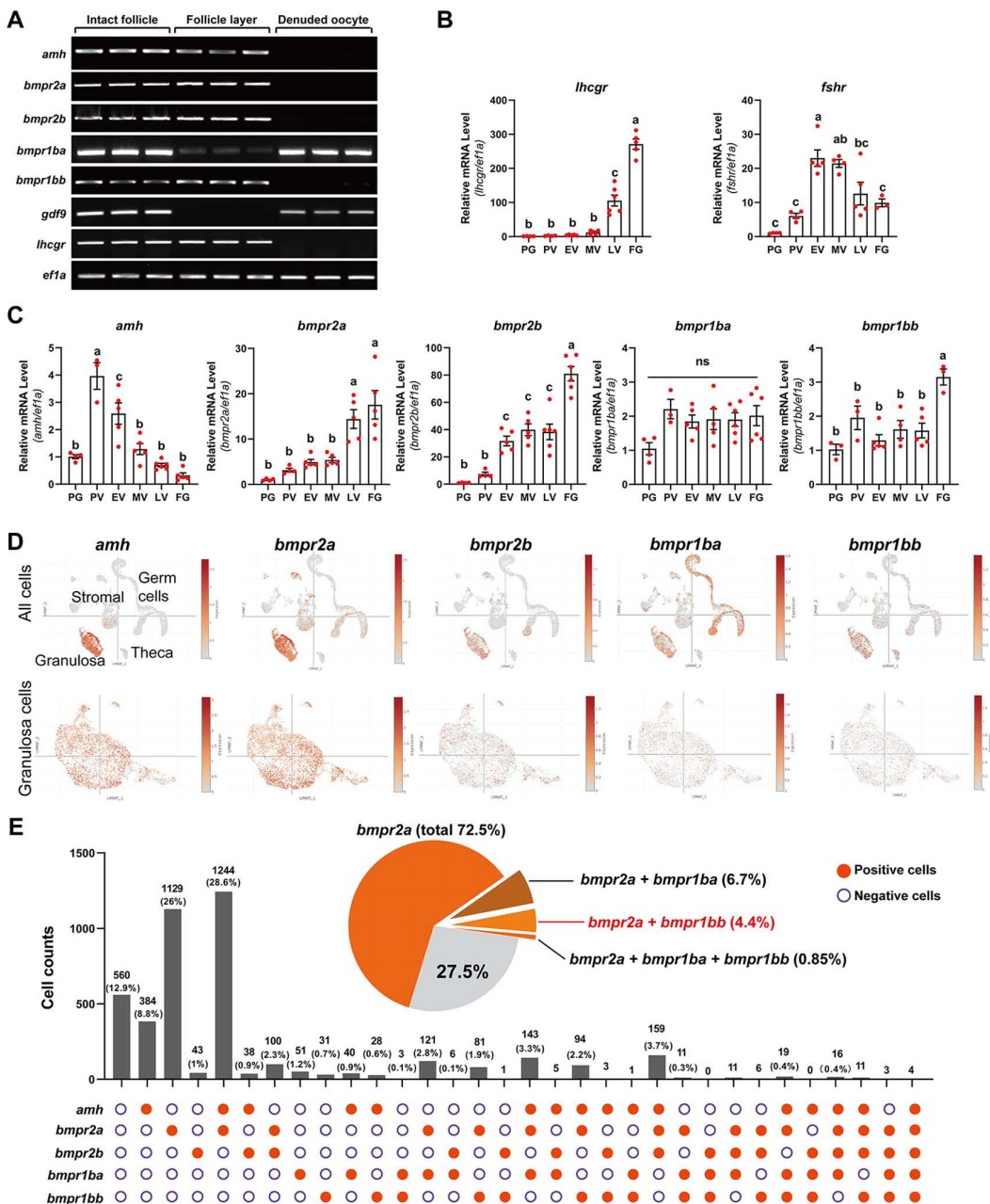

**Fig 1. Spatio-temporal expression profiles of anti-Müllerian hormone (*amh*) and BMP receptors (*bmpr2a, bmpr2b, bmpr1ba, bmpr1bb*) in the follicle. (A)** Intrafollicular distribution of *amh* and BMP receptors (*bmpr2a, bmpr2b, bmpr1ba, bmpr1bb*) in the FG follicle. The housekeeping gene *ef1a* serves as a reference, while *lhcgr* and *gdf9* are the marker genes expressed in the follicle layer and oocyte respectively. **(B-C)** Temporal expression profiles of *amh* and BMP receptors (*bmpr2a, bmpr2b, bmpr1ba, bmpr1bb*) during folliculogenesis. Quantitative assessment of mRNA levels was conducted using real-time qPCR, normalized to the expression of the housekeeping gene *ef1a*. The data are expressed as a fold change relative to the PG stage. Follicle staging was confirmed by examining the expression patterns of *lhcgr* and *fshr* as references. The values are mean ± SEM (n ≥ 3) from a representative experiment. Different letters indicate statistical significance (P < 0.05). PG, primary growth; PV, previtellogenic stage; EV, early vitellogenic stage; MV, mid-vitellogenic stage; LV, late vitellogenic stage; FG, full-grown. **(D)** Gene expression UMAP plots of *amh, bmpr2a, bmpr2b, bmpr1ba,* and *bmpr1bb*. Cells exhibiting expression of the specified gene are represented in orange, with varying shades indicating the levels of expression. The gradient of orange correlates with expression intensity, as detailed by the intensity scale on the right of each plot. The plots were all downloaded from the open-access Single Cell Portal (https://singlecell.broadinstitute.org/single_cell). **(E)** The colocalization of *amh* with BMP receptors (*bmpr2a, bmpr2b, bmpr1ba, bmpr1bb*) is observed in zebrafish granulosa cells, represented by histogram. The scRNA-seq raw data utilized for this analysis were sourced from the Gene Expression Omnibus (GEO) database (https://www.ncbi.nlm.nih.gov/geo/), under accession number GSE191137. The data was visualized using a custom R script, which was developed internally and incorporated Perl scripting.

transcriptome dataset on 40-dpf zebrafish ovaries [48], the developmental stage when gonad differentiation is almost complete [49,50]. This dataset allowed us to examine the spatial expression patterns of *amh* and its potential receptors in various follicle cell types. We obtained the single cell RNA-seq data from the Gene Expression Omnibus (GEO) database (accession number GSE191137) and analyzed gene expression patterns using UMAP plots from the Single Cell Portal (https://singlecell.broadinstitute.org/single_cell). Our analysis confirmed that *amh* and the BMP receptors examined (*bmpr2a*, *bmpr2b*, *bmpr1ba*, and *bmpr1bb*) were predominantly expressed in somatic cells, particularly in the granulosa cell lineage. Consistent with our RT-PCR results, *bmpr1ba* was more abundantly expressed in germ cells (Fig 1D). Focusing our analysis on the granulosa cells, we discovered some intriguing expression patterns, suggesting functional heterogeneity among granulosa cells.

First, *bmpr2a* was expressed in a large proportion of granulosa cells (72.5% in total), with 38.4% of these cells also co-expressing *amh*. Second, *bmpr2a* and *bmpr2b* exhibited distinct spatial expression patterns. Most of *bmpr2a*-expressing cells did not express *bmpr2b*, and only a small fraction of granulosa cells (7.1%) co-expressed both receptors. Third, we observed minimal co-expression between *bmpr1ba* and *bmpr1bb* (0.85%). In contrast, *bmpr1ba* co-expressed with *bmpr2a*, but not *bmpr2b,* in 6.7% of granulosa cells, and *bmpr1bb* showed a similar pattern, co-expressing with *bmpr2a* in 4.4% of cells. This expression pattern suggests potential existence of *bmpr2a-bmpr1ba* and *bmpr2a-bmpr1bb* signaling pathways within different granulosa cell subpopulations. Finally, the abundance of *bmpr2b*-expressing cells was much lower than that expressing *bmpr2a*, and they showed minimal co-expression with either *bmpr1ba* or *bmpr1bb* (1.4%), indicating that Bmpr2b signaling at this stage of follicle development may involve other BMP type I receptors, such as *bmpr1aa* and *bmpr1ab* (Fig 1E). Although the expression data are informative, we cannot completely rule out a role for Bmpr2b in Amh signaling without functional verification (see below).

## Mutagenesis of *bmpr1ba* and *bmpr1bb*

We previously proposed, based on genetic evidence, that Bmpr2a likely serves as the type II receptor for Amh in zebrafish [38]. An important next question is which type I receptor is involved in Amh-Bmpr2a signaling. A previous study reported that a point mutation in the BMP type I receptor *bmpr1bb* (also known as *alk6b*) (W256A) caused enlarged gonads with an accumulation of early germ cells in zebrafish [39]. This phenotype closely resembled those of *amh* and *bmpr2a* mutants, suggesting that these genes may function in a common signaling pathway to regulate germ cell development. Zebrafish possesses two paralogues of the Bmpr1b gene, *bmpr1ba* and *bmpr1bb*, which share high sequence identity (S1 Fig). Given this, we investigated the potential roles of both receptors in Amh-Bmpr2a signaling in the present study.

We first generated indel mutants for both *bmpr1ba* and *bmpr1bb* using the CRISPR/Cas9 method, as previously described [51]. For *bmpr1ba*, we established two mutant lines: one with a 5-bp deletion in exon 6 (ZFIN line number: umo71) and the other with a 34-bp insertion in the same exon (umo72). These mutant alleles are referred to as *bmpr1ba-/-(-5 bp)* and *bmpr1ba-/-(+34 bp)* respectively (S2A and S2B Fig).

Phenotype analysis of *bmpr1ba* mutants showed no difference between the two lines (-5 and +34) (S4A and S4B Fig); we therefore focused our phenotype analysis on *bmpr1ba-/-(-5 bp)* (referred to as *bmpr1ba-/-* in the present study). No apparent abnormalities in reproductive development or performance were observed in *bmpr1ba-/-*. At 100 dpf, *bmpr1ba-/-* females showed no significant differences from control fish (*bmpr1ba +/-;bmpr1bb +/-*) in terms of gross morphology (body size and color) (Fig 2A), ovarian size (gonadosomatic index, GSI) (Fig 2B), follicle composition (from PG to FG) (Fig 2C) and fecundity (number of eggs spawned each time) (Fig 2D). The male *bmpr1ba-/-* mutants also exhibited normal testis size with typical spermatogenesis compared to controls (Fig 3A and 3B).

For *bmpr1bb*, we also obtained two mutant lines with distinct deletions: *bmpr1bb-/-(-11)* and *bmpr1bb-/-(-20)*. The line *bmpr1bb-/-(-20)* harbored a 20-bp deletion in exon 1 (umo73), while *bmpr1bb-/-(-11)* had an 11-bp deletion in exon 11 (umo74) (S3A and S3B Fig). The *bmpr1bb-/-(-11 bp)* mutant produces a truncated protein containing the entire

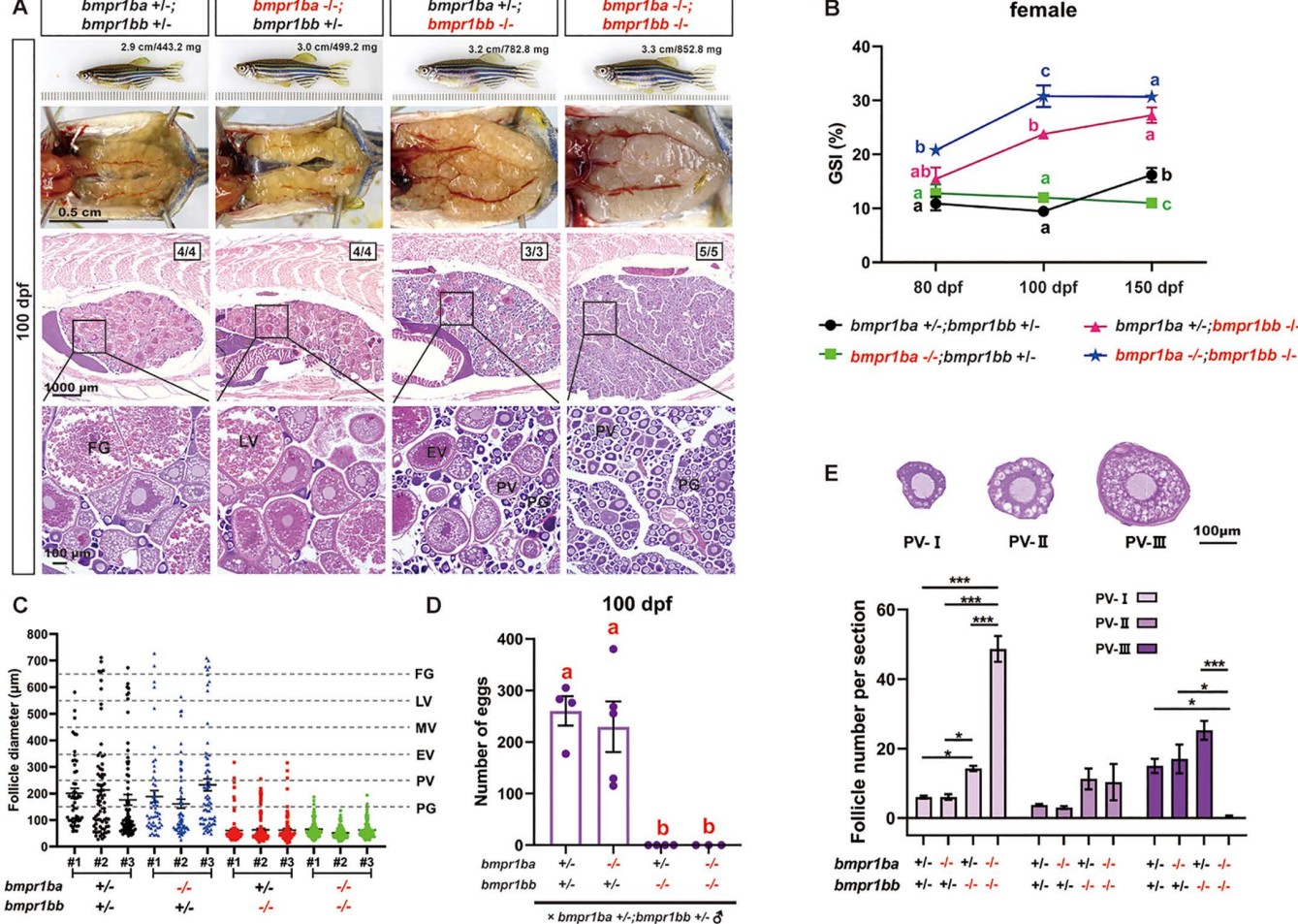

**Fig 2. The phenotype of BMP Type IB receptor mutations in females. (A)** Ovarian morphology and histology across genotypes at 100 dpf. Ovaries from both control and *bmpr1ba* single mutants showed normal growth and developmental patterns. In contrast, *bmpr1bb* single mutant and double mutants with *bmpr1ba* (*bmpr1ba-/-;bmpr1bb-/-*) exhibited a hypertrophic phenotype. The *bmpr1bb-/-* ovaries contained only PG to EV stage follicles, while double mutants contained primarily PG follicles with a few at early PV stage. The sizes of representative fish are shown on the top (cm, body length; mg, body weight), and the ratios in the photographs represent the number of samples exhibiting the displayed phenotype over the total number examined. **(B)** Gonadosomatic index (GSI) in different female genotypes at various developmental stages (80, 100, 150 dpf). Both *bmpr1bb* single and double mutants (*bmpr1ba-/-;bmpr1bb-/-*) exhibited significantly higher GSI values relative to the control group and *bmpr1ba* single mutant from 100 dpf (n ≥ 3). Statistical differences were analyzed using one-way ANOVA performed independently at each time point (80, 100, and 150 dpf). Different letters indicate significant differences (P < 0.05) between genotypes within the respective age group. **(C)** Follicle distribution across different genotypes in the ovary at 100 dpf. **(D)** Fertility in *bmpr1ba* and *bmpr1bb* single and double mutant females at 100 dpf. Each point represents the average number of eggs laid by three females in each breeding trial (n = 3-5 trials). **(E)** PV composition across *bmpr1ba* and *bmpr1bb* single and double mutants. The PV stage is categorized into three sub-stages based on the size and number of layers of cortical alveoli (CA). Specifically, PV-I is characterized by a single layer of small CAs; PV-II by a single layer of larger CAs; and PV-III by multiple layers of large CAs. Statistical significance is indicated by different letters (p < 0.05) or asterisks (* P < 0.05; **P < 0.01; ***P < 0.001). PG, primary growth; PV, pre-vitellogenic; EV, early vitellogenic; LV, late vitellogenic; FG, full-grown.

extracellular and transmembrane domains and a portion of the intracellular domain including the W256 site essential for receptor function, while the *bmpr1bb-/-(-20 bp)* mutant has lost almost the entire Bmpr1bb protein (S3B Fig).

Unlike *bmpr1ba* mutants, both *bmpr1bb* mutants, *bmpr1bb-/-(-11)* and *bmpr1bb-/-(-20)*, exhibited significant gonadal hypertrophy and accumulation of early-stage germ cells (PG follicles in the ovary and spermatogonia in the testis).

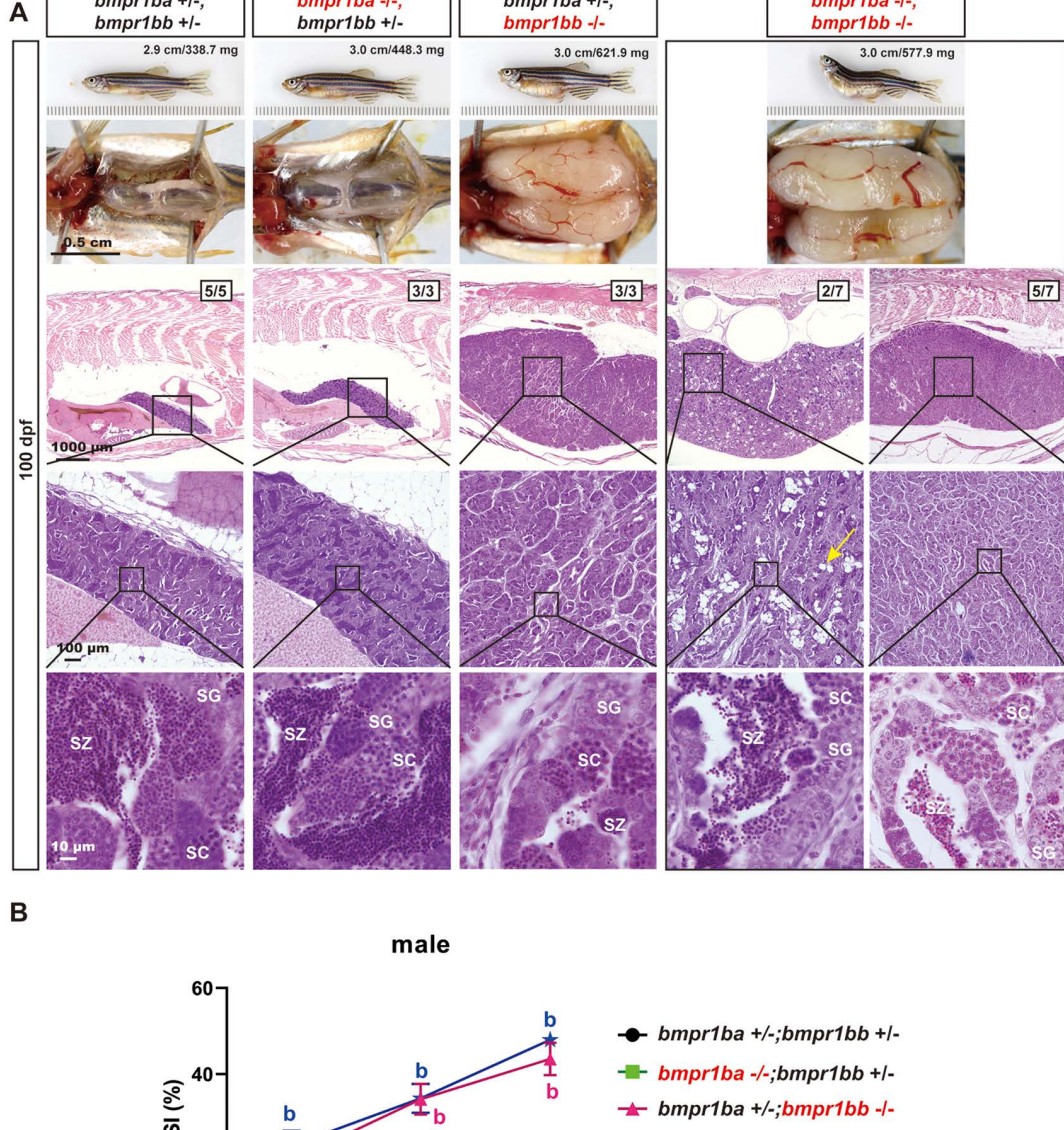

**Fig 3. Phenotypes of BMP Type IB receptor mutants (*bmpr1ba-/-* and *bmpr1bb-/-*) in males. (A)** Testicular morphology and histology across genotypes at 100 dpf. Testes from control and *bmpr1ba* single mutant (*bmpr1ba-/-*) exhibited normal growth and developmental patterns. In contrast, *bmpr1bb* single mutant (*bmpr1bb-/-*) and double mutants (*bmpr1ba-/-;bmpr1bb-/-*) displayed hypertrophic testes with limited spermatogenic activity, similar to that observed in *amh-/-* and *bmpr2a-/-* single mutants. Some double mutants displayed empty sacs in their testes (yellow arrows). **(B)** GSI across different genotypes at various developmental stages (80, 100, 150 dpf). Starting from 80 dpf, both *bmpr1bb* single and double mutants exhibited elevated GSI values relative to the control group and *bmpr1ba* single mutants (n ≥ 3). Statistical differences were analyzed using one-way ANOVA performed independently at each time point (80, 100, and 150 dpf). Different letters indicate statistical significance (P < 0.05) at each time point. SG, spermatogonia; SC, spermatocytes; SZ, spermatozoa.

Interestingly, while both *bmpr1bb* mutants showed similar phenotypes, the effects were more severe in *bmpr1bb*-/-(-11) (S5 Fig). By 180 dpf, *bmpr1bb*-/-(-11) females displayed ovarian hypertrophy with follicles predominantly at the PG stage (S5A Fig). In contrast, *bmpr1bb*-/-(-20) ovaries contained abundant follicles at advanced stages, including vitellogenic follicles, even at 210 dpf (S5B Fig). Similarly, both *bmpr1bb* mutant testes displayed severe hypertrophy, but their cellular characteristics differed. At 180 dpf, *bmpr1bb*-/-(-11) testes showed limited meiotic activity with numerous vacuoles and minimal mature spermatozoa, while *bmpr1bb*-/-(-20) testes lacked vacuoles and exhibited substantial meiosis, with mature spermatozoa present by 210 dpf (S5A and S5B Fig). There are two explanations for the observed discrepancy between the two mutants. First, the truncated form (-11) may function as a potential dominant negative mutant by sequestering shared type II receptors or ligands (S6 Fig), thereby inhibiting other BMP pathways beyond just Amh signaling. Second, the complete absence of Bmpr1bb in the *bmpr1bb* (−20) mutant may force type II Bmpr2a to recruit alternative and physiologically irrelevant TGF-β family type I receptors, thereby partially restoring Smad1/5/9 signaling. The exact mechanism is unknown, and it will be an interesting issue to explore in future studies. For the remainder of this study, we focused our analysis on *bmpr1bb*-/-(-11) mutant line, hereafter referred to as *bmpr1bb*-/-.

Similar to the *bmpr1ba*-/- mutant, the *bmpr1bb*-/- mutant showed no gross morphological abnormalities in either males or females (Fig 2A and 3A). In contrast to the severe growth defects observed in the double type II receptor mutants *bmpr2a*-/-;*bmpr2b*-/- (150 dpf), the *bmpr1ba*-/-;*bmpr1bb*-/- double mutants did not significantly impact somatic growth, as measured by body weight (BW) and body length (BL) (Fig 4A). Despite lack of impact on somatic growth, *bmpr1ba*-/-;*bmpr1bb*-/- double mutants exhibited more severe phenotype in gonads especially folliculogenesis in the ovary, containing predominantly PG follicles (Fig 4B, 4C and 4D). This observation suggests that while Bmpr1ba plays a limited role in Amh signaling, it may act in a complementary manner with Bmpr1bb.

## Genetic evidence for Amh-Bmpr2a-Bmpr1bb signaling pathway

In contrast to *bmpr1ba*-/-, *bmpr1bb*-/- mutants (-11) exhibited severe gonadal hypertrophy in both sexes at 100 dpf, phenocopying the mutants of *bmpr2a*-/- [38] and *amh*-/- [34]. In females, the *bmpr1bb*-/- single mutants displayed significant ovarian hypertrophy as shown by obviously enlarged ovary (Fig 2A) and significantly increased GSI (Fig 2B). Histological examination and follicle composition analysis demonstrated an abundance of PG follicles in the mutant ovary with some follicles growing maximally to EV stage (Fig 2A and 2C). These mutant females could not spawn with males at 100 dpf, therefore being infertile at this age (Fig 2D). Interestingly, double mutation of *bmpr1ba* and *bmpr1bb* seemed to cause more severe phenotypes. The double mutant ovaries (*bmpr1ba*-/-;*bmpr1bb*-/-) were even larger than those of *bmpr1bb* single mutant, reaching the peak size at 100 dpf (Fig 2A and 2B). The follicles in the double mutant ovaries were predominantly at PG stage with some transitioned to PV stage; however, most PV follicles were stalled at early PV (PV-I) and no late PV follicles (PV-III) were observed at 100 dpf (Fig 2A, 2C and 2E). The phenotypes of *bmpr1bb*-/- females became progressively more severe with aging. At 150 dpf, *bmpr1bb*-/- ovaries were closer in size to those in the double mutants (*bmpr1ba*-/-;*bmpr1bb*-/-) and contained predominantly PG follicles. Some follicles could undergo PG-PV transition to reach PV-I; however, no vitellogenic follicles were observed, in contrast to the ovaries at 100 dpf. Meanwhile, the ovaries of double mutants contained PG follicles only, also presenting a more severe phenotype than that at 100 dpf (Fig 4B and 4D).

The testes of *bmpr1bb*-/- mutants were significantly enlarged as shown by gross morphology (Fig 3A) and GSI at 80, 100 and 150 dpf (Fig 3B). Histological analysis at 100 dpf revealed that the enlarged testes were characterized by an accumulation of spermatogonia and limited mature spermatozoa. Double mutations of *bmpr1ba* and *bmpr1bb* produced no additional phenotypes compared to *bmpr1bb*-/- mutants, except that the testes of two double mutants contained abundant vacuoles (Fig 3A and 3B).

Our results suggested the existence and functional importance of the Amh-Bmpr2a-Bmpr1bb signaling pathway in controlling gonadal homeostasis in both male and female zebrafish. This proposal is based on the observation that *bmpr1bb*-/-, but not *bmpr1ba*-/-, could phenocopy the mutants of *amh*-/- and *bmpr2a*-/- in both males and females (Fig 4B and 4C). Interestingly,

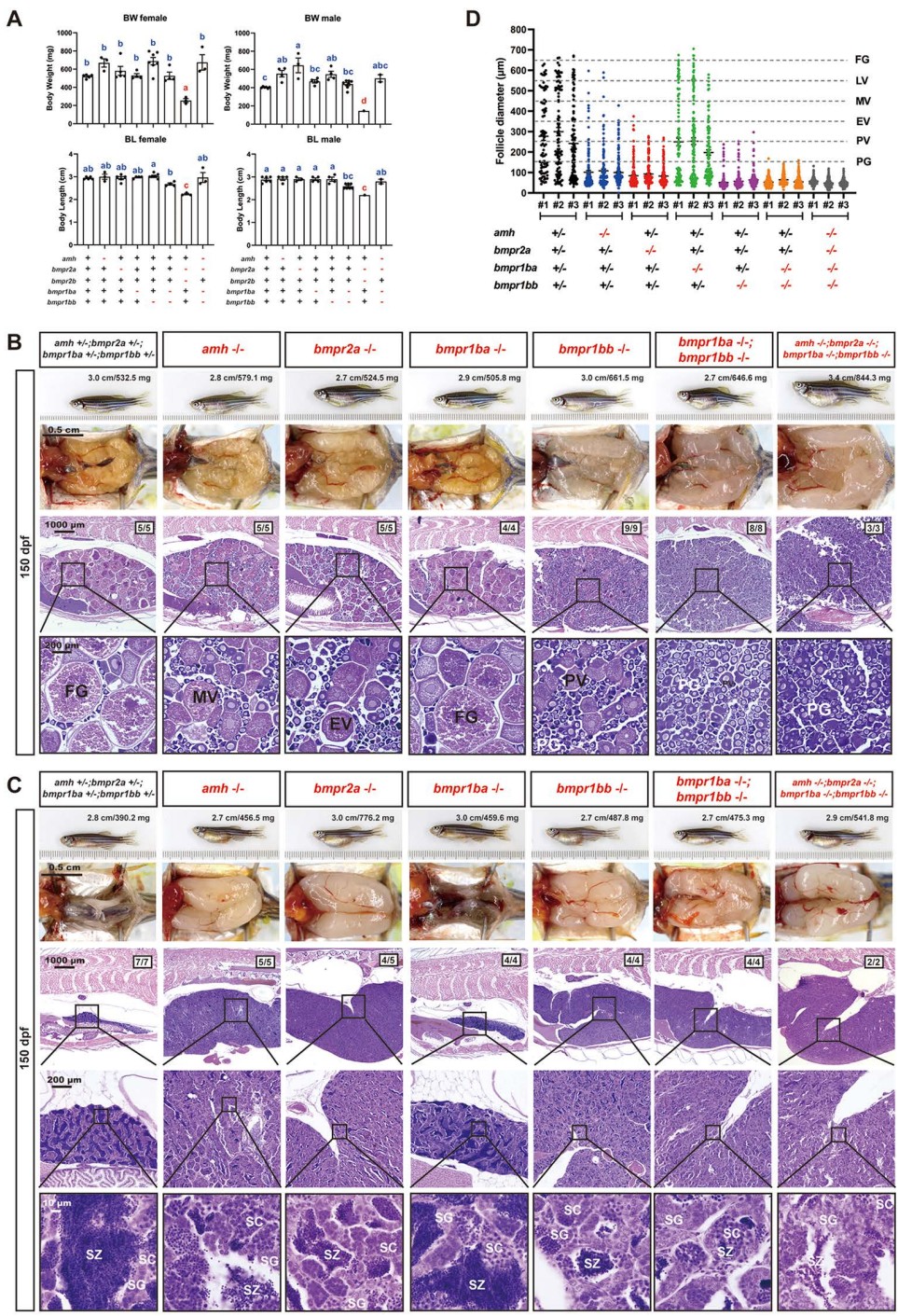

**Fig 4. Histological analysis of *amh, bmpr2a, bmpr1ba, bmpr1bb* single and quadruple mutants (*amh-/-;bmpr2a-/-;bmpr1ba-/-;bmpr1bb-/-*) at 150 dpf. (A)** Body weight (BW) and standard body length (BL) across various genotypes in females and males. **(B)** Ovarian morphology and histology across genotypes at 150 dpf. **(C)** Testicular morphology and histology across genotypes at 150 dpf. **(D)** Follicle distribution across different genotypes in the ovary. Different letters indicate statistical significance (p < 0.05). PG, primary growth; PV, pre-vitellogenic; EV, early vitellogenic; MV, mid-vitellogenic; FG, full-grown; SG, spermatogonia; SC, spermatocytes; SZ, spermatozoa.

mutation within the Amh-Bmpr2a-Bmpr1bb pathway exhibited a gradient of phenotypic severity, particularly in the ovary, with *bmpr1bb-/-* displaying the most severe effects, followed by *bmpr2a-/-* and then *amh-/-* (*bmpr1bb-/- > bmpr2a-/- > amh-/-*). Simultaneous loss of all members of the pathway together with *bmpr1ba-/-* in the quadruple mutants (*amh-/-;bmpr2a-/-; bmpr1ba-/-;bmpr1bb-/-*) did not yield any additional phenotypes compared to the *bmpr1ba-/-;bmpr1bb-/-* double mutants in both ovary and testis (Fig 4B-4D).

## Biochemical evidence for Amh-Bmpr2a-Bmpr1bb signaling pathway

To further demonstrate the role of Bmpr2a as the type II receptor for Amh in zebrafish, we examined Smad1/5/9 phosphorylation in response to recombinant zebrafish Amh in cultured follicle cells from WT, *bmpr2a-/-* and *bmpr2b-/-* fish. The recombinant Amh (zf-Amh) was produced from a stable CHO cell line (Fig 5A). Treatment of the cultured follicle cells, which express both *bmpr2a* and *bmpr2b,* with concentrated conditioned medium from the recombinant cell line producing zf-Amh induced a time- and dose-dependent increase in Smad1/5/9 phosphorylation in WT follicle cells, with 100 µL/mL for 30 min inducing a maximal response (Fig 5B and 5C). In contrast to the control follicle cells that exhibited a statistically

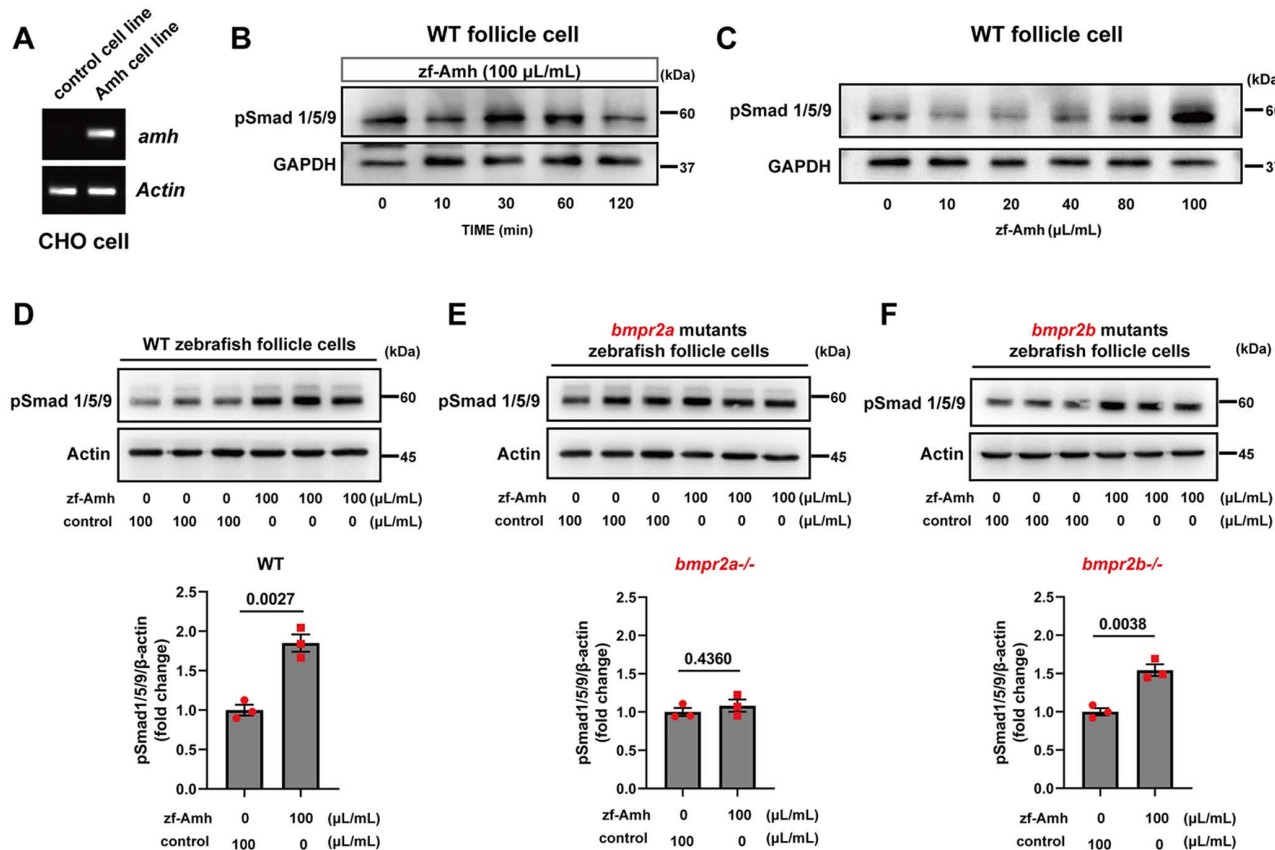

**Fig 5. Biochemical evidence for Amh-Bmpr2a-Bmpr1bb signaling in vitro. (A)** Expression of Zebrafish Amh in Amh-Flp-In CHO Cells. CHO cells of the Flp-In Complete System were employed to create a cell line expressing zebrafish Amh. RT-PCR was utilized to verify zebrafish Amh expression in the Amh-Flp-In CHO cells. **(B)** Time course of Smad1/5/9 phosphorylation response to recombinant zebrafish Amh (zf-Amh; 100 µL/mL) in follicle cells after incubation for 0, 10, 30, 60, 120 min. **(C)** Dose-dependent effects of zf-Amh on Smad1/5/9 phosphorylation at 30 min in follicle cells. The medium containing zf-Amh was combined with varying volumes of control medium, totaling 100 µL in each well. **(D-F)** Influence of zf-Amh on Smad1/5/9 phosphorylation in WT, *bmpr2a-/-* and *bmpr2b-/-* follicle cells, respectively. The P values for statistical significance are shown on top of the graphs (n = 3).

significant response to zf-Amh (Fig 5D), the responsiveness was completely abolished in the *bmpr2a-/-* follicle cells (Fig 5E) but not *bmpr2b-/-* cells (Fig 5F), suggesting the involvement of Bmpr2a, but not Bmpr2b, in Amh signaling.

## Functional evidence for Amh-Bmpr2a-Bmpr1bb signaling pathway

We suggested in our previous study that Amh plays an important role in controlling gonadal homeostasis in both females and males by inhibiting gonadotropin signaling. This was supported by our finding that simultaneous mutations of *amh* and gonadotropin receptors (*fshr* and *lhcgr*) completely abolish the phenotype of gonadal hypertrophy in both females (*amh-/-;fshr-/-*) and males (*amh-/-;fshr-/-;lhcgr-/-*) [38]. To provide functional evidence for participation of Bmpr2a and Bmpr1bb in Amh-Bmpr2a-Bmpr1bb signaling pathway, we adopted a similar approach to examine if the phenotypes of *bmpr2a* and *bmpr1bb* mutants were also dependent on gonadotropin signaling.

As reported previously on *amh* mutant [34], we generated double knockout females (*bmpr2a-/-;fshr-/-*) and triple knock-out males (*bmpr2a-/-;fshr-/-;lhcgr-/-*). Phenotypic analysis at 90 dpf revealed that the *bmpr2a-/-* mutant females displayed ovarian hypertrophy with significantly more abundant PG follicles, similar to *amh* mutant [34]. In contrast, the *fshr-/-* females exhibited severe ovarian hypotrophy with only a few early PG follicles, consistent with our previous findings [52]. Interestingly, simultaneous mutations of *bmpr2a* and *fshr* abolished the hypertrophic phenotype of *bmpr2a-/-* in the double mutant females (*bmpr2a-/-;fshr-/-*), resulting in extremely underdeveloped ovaries containing a small number of early PG follicles. It is worth noting that some double mutants (3/5) began to undergo sex reversal, developing testicular tissues alongside ovarian follicles (Fig 6A). This observation is similar to the phenotype of *amh* and *fshr* double mutants (*amh-/-;fshr-/-*) [34].

In males, the two gonadotropin receptors (*fshr* and *lhcgr*) show strong functional complementarity [52], necessitating simultaneous mutations of both receptors to block gonadotropin signaling. To test their roles in Bmpr2a signaling in males, we created a triple knockout involving both *fshr* and *lhcgr* (*bmpr2a-/-;fshr-/-;lhcgr-/-*). Similar to the *amh* mutant, the single *bmpr2a* mutant also resulted in severe testis hypertrophy with reduced spermatogenic activity. In contrast, the loss of both gonadotropin receptors in the double mutants (*fshr-/-;lhcgr-/-*) caused severe testis hypotrophy with limited meiotic activity and sperm production. Simultaneous mutation of all three genes (*bmpr2a-/-;fshr-/-;lhcgr-/-*) resulted in a complete loss of the hypertrophic phenotype observed in *bmpr2a-/-* males, with no meiotic activity (Fig 6B). This outcome agrees well with the phenotype of the *amh* mutant in the absence of two gonadotropin receptors [34].

We next investigated the role of Bmpr1bb in gonadotropin signaling using a similar approach. Loss of *bmpr1bb* in females resulted in ovarian hypertrophy characterized by a disproportionate increase in the number of PG follicles. This phenotype could be completely prevented by simultaneous mutation of *fshr* knockout (*bmpr1bb-/-;fshr-/-*) at 90 dpf (Fig 7A). Similarly, the severe testis hypertrophy observed in *bmpr1bb-/-* males was also abolished by the simultaneous loss of both *fshr* and *lhcgr* (*bmpr1bb-/-;fshr-/-;lhcgr-/-*). It is worth noting that while the double mutants of *fshr* and *lhcgr* in the presence of *bmpr1bb* (*bmpr1bb+/-;fshr-/-;lhcgr-/-*) exhibited limited meiotic activity in the testis at 90 dpf, the triple mutant males (*bmpr1bb-/-;fshr-/-;lhcgr-/-*) completely lacked this activity (Fig 7B).

Since double mutation of *bmpr1ba* and *bmpr1bb* mimicked the phenotype of *bmpr1bb* mutant with increased severity, especially in females, we also tested *bmpr1ba* in the analysis. We generated a triple knockout (*bmpr1ba-/-;bmpr1bb-/-;fshr-/-*) for females and a quadruple knockout (*bmpr1ba-/-;bmpr1bb-/-;fshr-/-;lhcgr-/-*) for males. Consistent with *amh, bmpr2a* and *bmpr1bb* single mutants, the loss of *fshr* completely abolished the ovarian hypertrophy in Bmpr1b double mutants (*bmpr1ba-/-;bmpr1bb-/-*) at 90 dpf (S7A Fig), and simultaneous loss of *fshr* and *lhcgr* also abolished the severe testis hypertrophy shown by Bmpr1b double mutants (*bmpr1ba-/-;bmpr1bb-/-*) (S7B Fig).

These results suggest that the Amh-Bmpr2a-Bmpr1bb pathway may work by suppressing gonadotropin signaling. Dysfunction of any component within this pathway could lead to enhanced gonadotropin signaling activity, resulting in hypertrophic gonadal growth. To provide further support for this regulatory mechanism, we analyzed the spatial expression patterns of *fshr, lhcgr, bmpr2a* and *bmpr2b* at single-cell resolution using the same dataset described previously [48]. *Fshr*

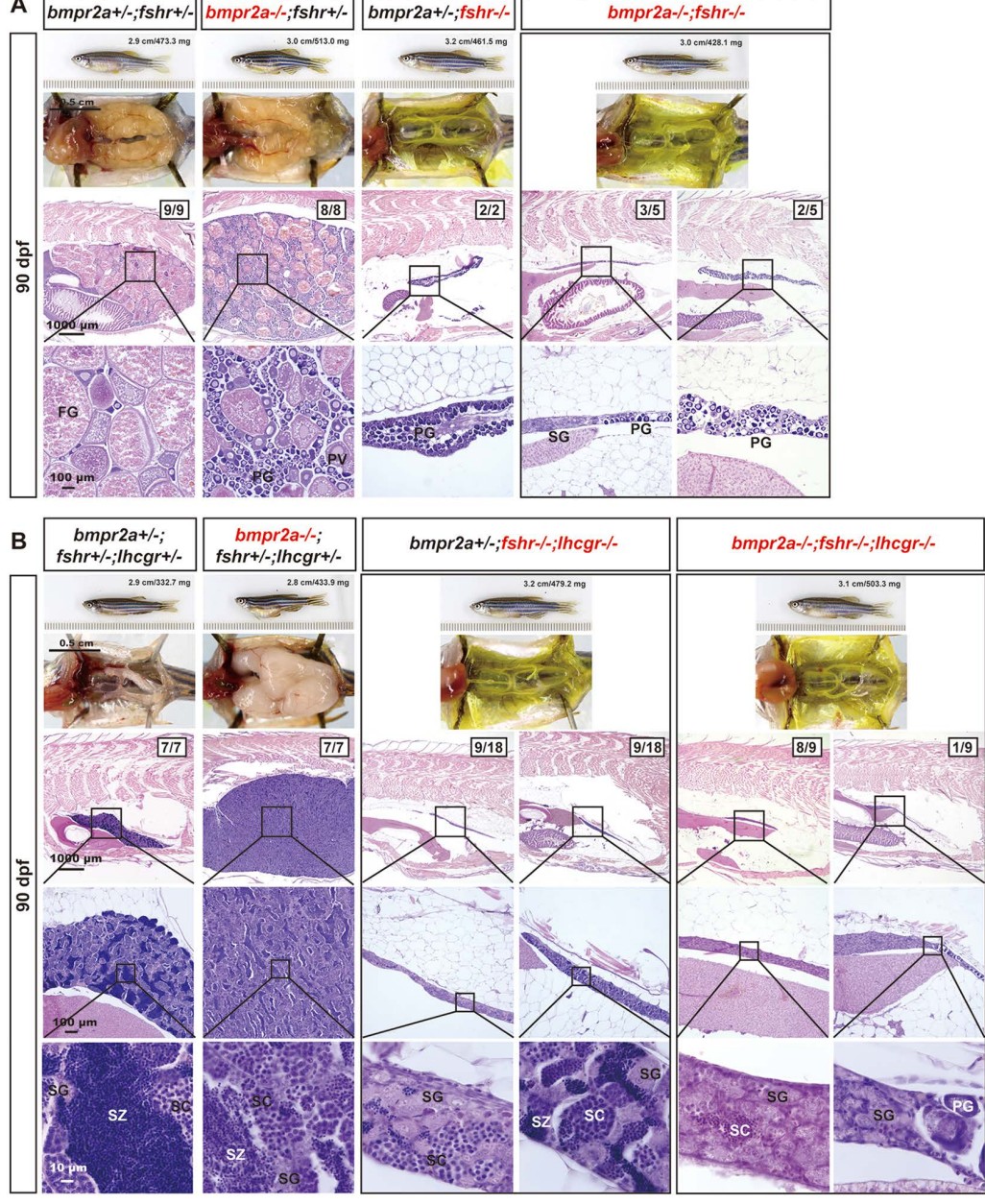

**Fig 6. Evidence for involvement of gonadotropin signaling in *bmpr2a-/-* phenotype. (A)** Ovarian morphology and histology of different genotypes at 90 dpf. The *bmpr2a* mutation alone (*bmpr2a-/-;fshr+/-*) led to ovarian hypertrophy, characterized by an abundance of early-stage follicles (PG and PV), while the *fshr* mutation (*bmpr2a+/-;fshr-/-*) resulted in ovarian hypotrophy with only early PG follicles present. Double mutation in both *bmpr2a* and *fshr* (*bmpr2a-/-;fshr-/-*) rescued the hypertrophic phenotype observed in the *bmpr2a* single mutant, with only PG follicles in the ovaries. Some individuals (3/5) showed signs of sex reversal with testicular tissues (mostly SG) co-existing with ovarian tissues. **(B)** Testis morphology and histology of different genotypes at 90 dpf. The *bmpr2a* mutation (*bmpr2a-/-;fshr+/-;lhcgr+/-*) caused testicular hypertrophy and disrupted spermatogenesis, with a reduction in meiosis. The double mutants for *fshr* and *lhcgr* (*bmpr2a+/-;fshr-/-;lhcgr-/-*) resulted in severe testicular hypotrophy; half of these samples displayed dysfunctional spermatogenesis with little meiotic activity, while the others had minimal amount of spermatozoa (SZ). Loss of gonadotropin receptors eliminated the testicular hypertrophy displayed by the *bmpr2a* mutant in the triple knockout (*bmpr2a-/-;fshr-/-;lhcgr-/-*), with no meiotic activity and complete absence of SZ. PG, primary growth; PV, pre-vitellogenic; FG, full-grown; SG, spermatogonia; SC, spermatocytes; SZ, spermatozoa.

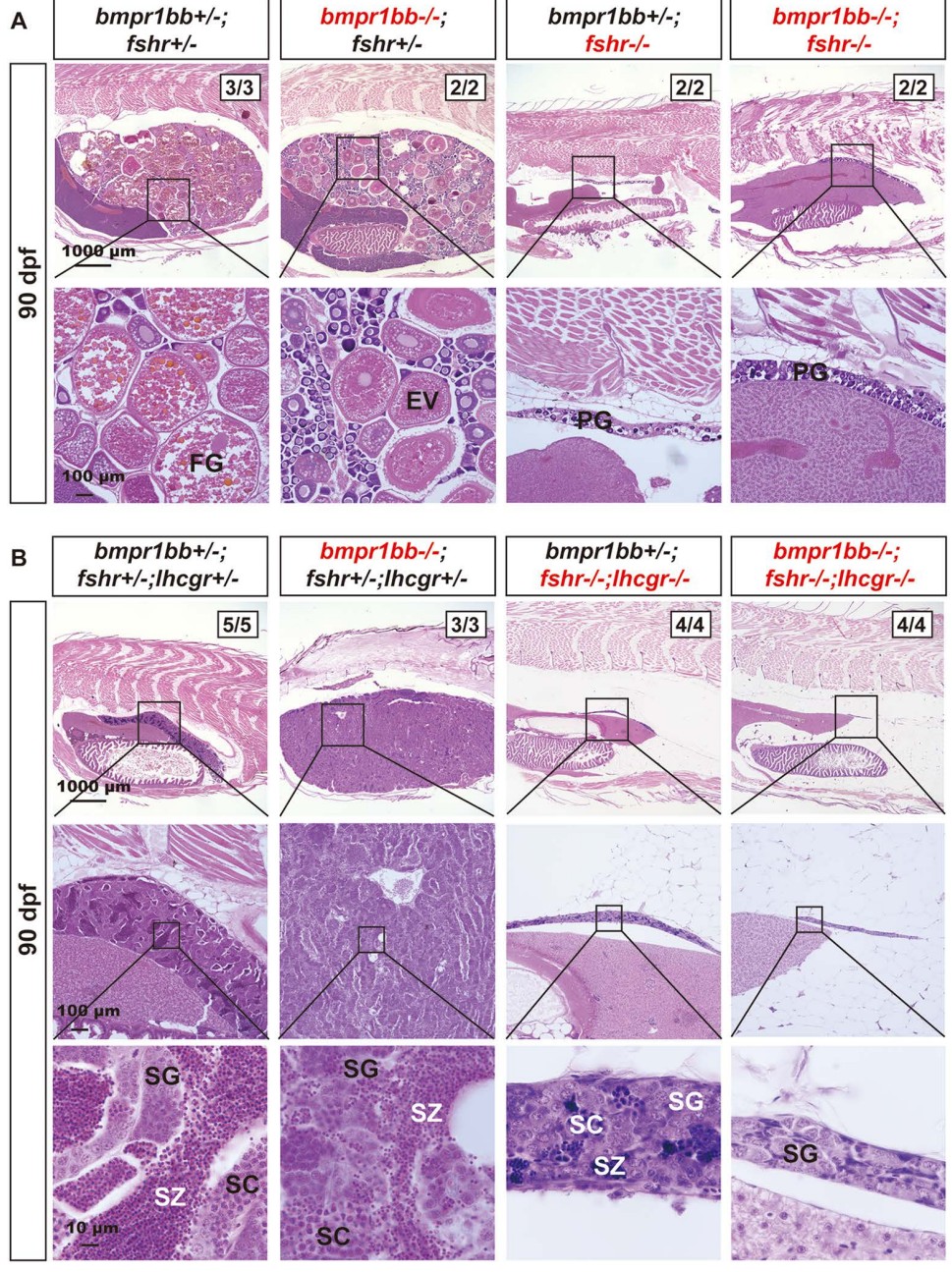

**Fig 7. Evidence for involvement of gonadotropin signaling in *bmpr1bb-/-* phenotype. (A)** Ovarian morphology and histology of different genotypes at 90 dpf. The *bmpr1bb* mutation (*bmpr1bb-/-;fshr+/-*) induced ovarian hypertrophy, characterized by the accumulation of early-stage follicles, while the *fshr* mutation (*bmpr1bb+/-;fshr-/-*) resulted in ovarian hypotrophy with only early PG follicles. Double mutation (*bmpr1bb-/-;fshr-/-*) rescued the hypertrophic phenotype displayed by the *bmpr1bb* mutant, with only PG follicles in the ovaries. **(B)** Testis morphology and histology of different genotypes at 90 dpf. The *bmpr1bb* mutation (*bmpr1bb-/-;fshr+/-;lhcgr+/-*) caused testicular hypertrophy and disrupted spermatogenesis with reduced meiosis. Double mutation of *fshr* and *lhcgr* (*bmpr1bb+/-;fshr-/-;lhcgr-/-*) resulted in testicular hypotrophy with minimal SZ. The hypertrophic testis growth in *bmpr1bb-/-* was eliminated in the triple knockout without *fshr* and *lhcgr* (*bmpr1bb-/-;fshr-/-;lhcgr-/-*), which showed little meiotic activity. PG, primary growth; EV, early vitellogenic; FG, full-grown; SG, spermatogonia; SC, spermatocytes; SZ, spermatozoa.

was expressed in both granulosa and theca cells, while *lhcgr* expression was mainly restricted to theca cells with some cells also co-expressing *fshr*. Interestingly, the majority (~77%) of *fshr*-positive granulosa cells, but not theca cells, also co-expressed *bmpr2a* (Fig 8A). In contrast, few *lhcgr*-positive theca cells co-expressed *bmpr2a* or *bmpr2b* (Fig 8B). This finding strongly suggests that the Amh-Bmpr2a-Bmpr1bb pathway specifically regulates Fshr signaling, rather than Lhcgr signaling, in the ovary.

### Roles of FSH and LH in Amh deficiency-induced gonadal hypertrophy

Our data from previous [34,38] and present studies demonstrated that mutations of *amh*, *bmpr2a* and *bmpr1bb* all induced gonadal hypertrophy with disproportionately abundant PG follicles and spermatogonia in females and males, respectively, and that the hypertrophic phenotypes of all three mutants could be prevented by simultaneous mutation of gonadotropin receptors (*fshr* and *lhcgr*). In zebrafish, gonadotropin signaling involves two canonical pathways (FSH-Fshr and LH-Lhcgr) and one non-canonical pathway (LH-Fshr) [52,53]. Although our studies, including this one, have established the

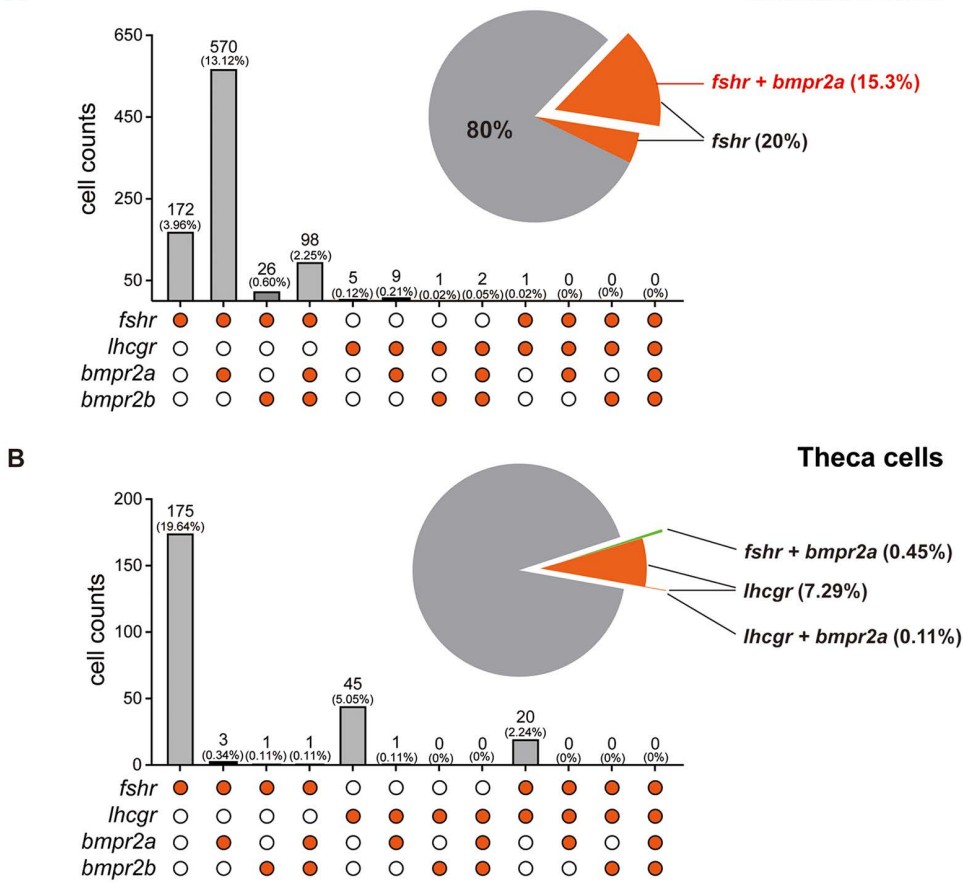

**Fig 8. Spatial expression patterns of *fshr*, *lhcgr*, *bmpr2a* and *bmpr2b* in granulosa and theca cells. (A)** Distribution of *fshr*, *lhcgr*, *bmpr2a* and *bmpr2b* expression in granulosa cells. **(B)** Distribution of *fshr*, *lhcgr*, *bmpr2a* and *bmpr2b* expression in theca cells. The scRNA-seq data utilized for this analysis were obtained from a GEO database under accession number GSE191137. The solid and empty circles indicate presence and absence of expression, respectively.

importance of gonadotropin signaling in the gonadal hypertrophy seen in *amh*, *bmpr2a*, and *bmpr1bb* mutants, the specific roles of FSH and LH in this process remained unclear. To further elucidate this, we generated a series of double and triple mutants involving *amh*, *fshb* and *lhb*.

At 90 dpf, all triple mutants examined (*amh-/-;fshb-/-;lhb-/-*, n = 3) developed as males, but with hypotrophic testes containing only spermatogonia (SG) and no apparent meiotic activity, contrasting with the control and testicular hypertrophy seen in *amh-/-* single mutant (Fig 9A and 9B). Surprisingly, at 240 dpf, testicular hypertrophy occurred in the triple mutants (*amh-/-;fshb-/-;lhb-/-*) despite the absence of both FSH and LH (Fig 9A), in contrast to *fshb* and *lhb* double mutants (*amh +/-;fshb-/-;lhb-/-*) (Fig 9C). Reintroducing either *fshb* or *lhb* in the combination (*amh-/-;fshb +/-;lhb-/-* and *amh-/-;fshb-/-;lhb +/-*) resulted in significant testicular hypertrophy at 90 dpf, with increased pre-meiotic spermatogonia but decreased meiotic spermatocytes and post-meiotic mature spermatozoa (Fig 9D and 9E). However, reintroducing *amh* (*amh +/-;fshb-/-;lhb-/-*) significantly increased meiotic activity with abundant spermatocytes (SC) and mature spermatozoa (SZ) in the testis at 90 dpf (Fig 9C) compared to the age-matched triple mutants, which contained SG only (Fig 9A). The testes of *amh +/-;fshb-/-;lhb-/-* fish showed normal growth and development at 240 dpf with all stages of spermatogenic cells present, indicating active spermatogenesis (Fig 9C). Surprisingly, in contrast to our previous report that the *fshb* and *lhb* double mutants (*fshb-/-;lhb-/-*) were all males at 100–120 dpf (31/31) [54], we identified two female individuals at 240 dpf in triple mutants *amh-/-;fshb-/-;lhb-/-* (2/8) and two in double mutants *amh +/-;fshb-/-;lhb-/-* (2/13), respectively. The ovaries of all four females were underdeveloped with follicles arrested completely at early PG stage, consistent with the phenotype observed in FSH receptor mutant *fshr-/-* [52], while displaying a greater number of PG follicles (Fig 9A and 9C).

In females, the presence of FSH (*amh-/-;fshb +/-;lhb-/-*) led to ovarian hypertrophy at 90 dpf, with GSI significantly higher than that of the age-matched control fish. The ovaries underwent normal folliculogenesis with all stages of follicles present from PG to FG stage. By comparison, the presence of LH (*amh-/-;fshb-/-;lhb +/-*) did not cause ovarian hypertrophy at 90 dpf, and ovaries were even smaller than those in the control fish despite normal folliculogenesis. This is in sharp contrast to the hypertrophic testes in males (*amh-/-;fshb-/-;lhb +/-*) (Fig 10A and 10B). At 240 dpf, however, ovarian hypertrophy occurred in both *amh-/-;fshb +/-;lhb-/-* (with FSH) and *amh-/-;fshb-/-;lhb +/-* (with LH). However, folliculogenesis ceased in both mutant ovaries, which accumulated large number of PG follicles with some early PV follicles occasionally observed (Fig 10A and 10B).

In addition to morphological and histological examination, we also determined the serum levels of 11-ketotestosterone (11-KT) in males with different *amh*, *fshb* and *lhb* mutations at 240 dpf. As shown in Fig 10C, the 11-KT levels in all double and triple mutants were significantly lower than that in the control fish, with the triple mutants showing the lowest levels (Fig 10C). We further analyzed the expression of genes involved in androgen synthesis by real-time qPCR at 270 dpf, including *star, cyp11a1, cyp11c1, cyp17a1, hsd3b1, hsd11b2, hsd17b1,* and *hsd17b3*. All the genes examined, except *cyp11a1,* showed a significant decrease in expression in all double (*amh +/-;fshb-/-;lhb-/-, amh-/-;fshb +/-;lhb-/-, and amh-/-;fshb-/-;lhb +/-*) and triple (*amh-/-;fshb-/-;lhb-/-*) mutants, especially in the absence of *amh*. The lowest levels were observed in the triple mutant (Fig 10D).

## Discussion

AMH is a member of the TGF-β superfamily that signals through a specific type II receptor (AMHRII) in mammals and teleosts like medaka and tilapia. Surprisingly, zebrafish also has Amh but lacks the specific type II receptor in its genome. How Amh signals in zebrafish has been an intriguing question. Building on our previous work [38] and others [39], this study provides compelling evidence that Amh signals through a novel pathway in zebrafish utilizing Bmpr2a as type II and Bmpr1bb as type I receptors. Our data rule out the participation of paralogues Bmpr2b and Bmpr1ba and demonstrate an intricate interplay between Amh and gonadotropins in regulating gonadal homeostasis.

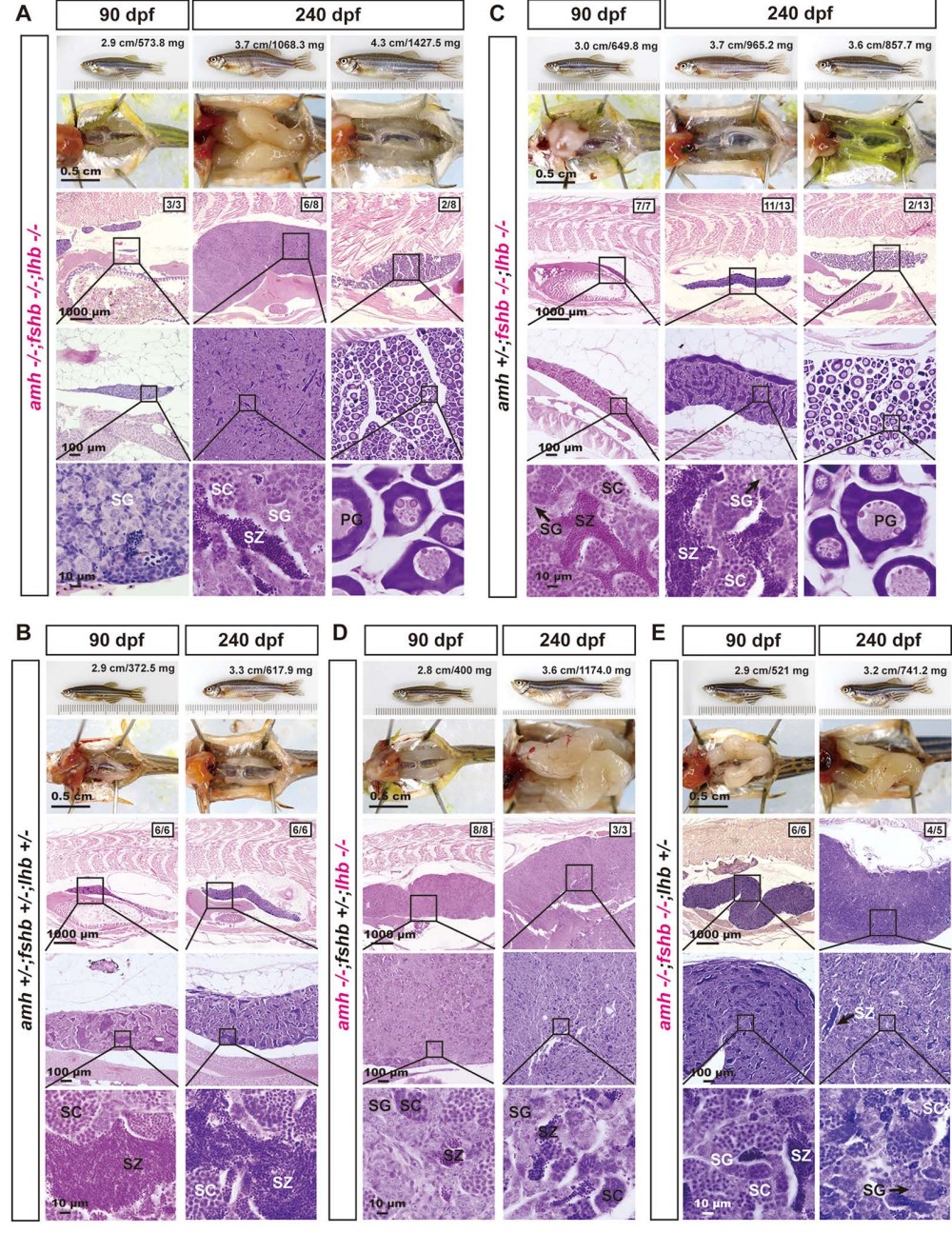

**Fig 9. Roles of gonadotropins FSH and LH in the phenotypic expression of *amh-/-* males. (A)** Triple mutants of *amh*, *fshb* and *lhb* (*amh-/-;fshb-/-;lhb-/-*) at 90 and 240 dpf. Three fish examined at 90 dpf were all males, which showed no testicular hypertrophy with SG only in the testis. However, testicular hypertrophy appeared at 240 dpf. Two females were identified at 240 dpf (2/8), showing ovarian hypertrophy with PG follicles only. **(B)** Age-matched control male fish (*amh+/-;fshb+/-;lhb+/-*). The testes contained all stages of spermatogenic cells with abundant mature spermatozoa. **(C-E)** Double mutants with one of the three genes (*amh*, *fshb* and *lhb*) present. Two females were identified in *fshb-/-;lhb-/-* double mutants with all follicles blocked at PG stage. Restoring either *fshb* or *lhb* in the triple mutants (*amh-/-;fshb+/-;lhb-/-* or *amh-/-;fshb-/-;lhb+/-*) led to severe testicular hypertrophy at both 90 and 240 dpf. The testes contained predominantly spermatogonia with limited spermatogenic activity. PG, primary growth; SG, spermatogonia; SC, spermatocytes; SZ, spermatozoa.

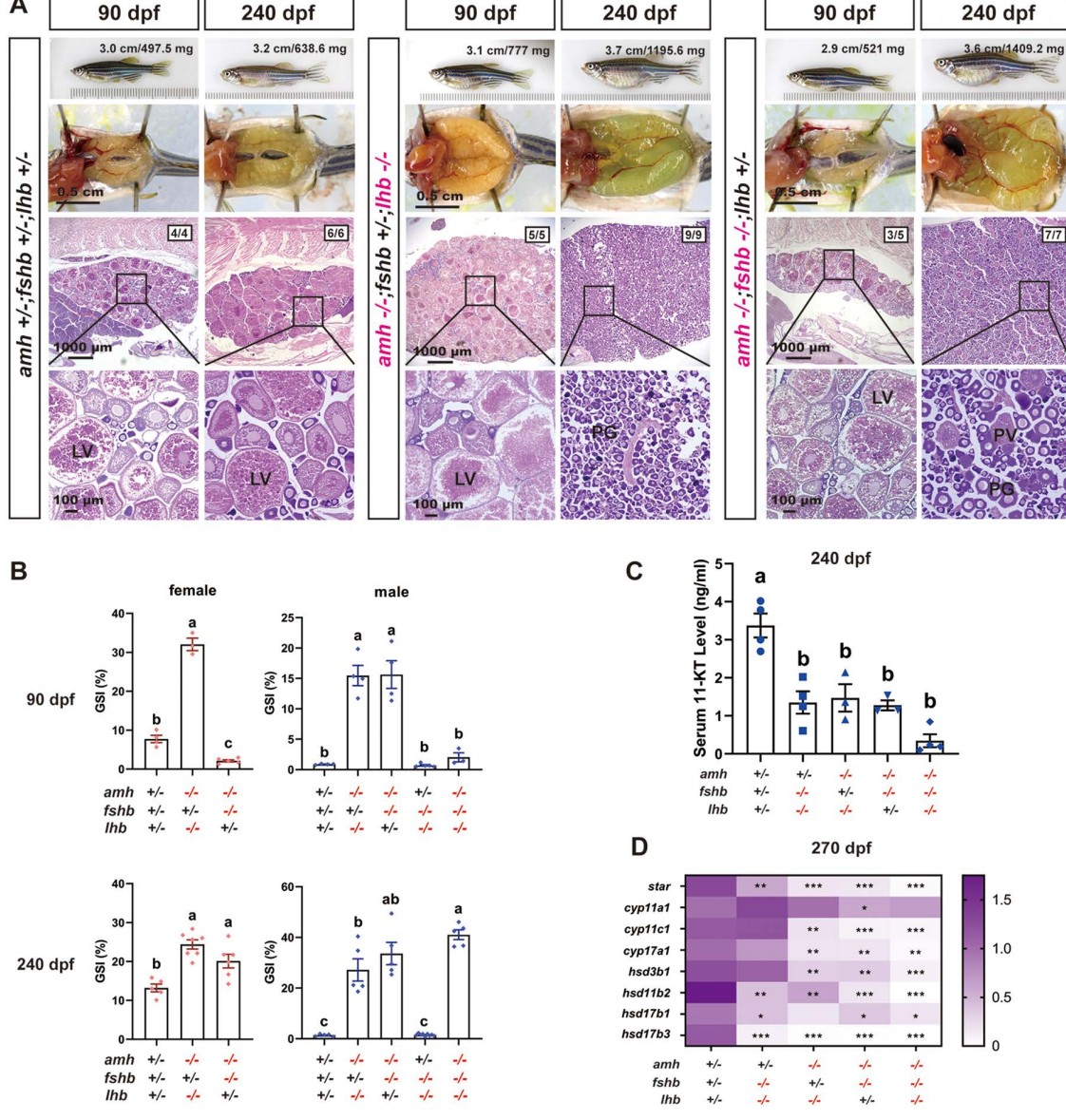

**Fig 10. Roles of gonadotropins FSH and LH in the phenotypic expression of *amh-/-* females. (A)** Ovarian morphology and histology at 90 and 240 dpf across genotypes. Restoring *fshb* in triple mutants (*amh-/-;fshb +/-;lhb-/-*) led to distinct results at different stages. The ovary exhibited hypertrophy at 90 dpf but maintained normal vitellogenic growth. However, the hypertrophy intensified at 240 dpf with ovaries containing PG follicles only. In the presence of *lhb* (*amh-/-;fshb-/-;lhb +/-*), there was a noticeable delay in ovarian development at 90 dpf, with the ovaries being smaller than those of the age-matched control group. However, the ovary displayed hypertrophy at 240 dpf, containing early stages of follicles (PG and early PV). **(B)** GSI across different genotypes at 90 and 240 dpf in males and females. **(C)** Measurement of 11-ketotestosterone (11-KT) in male zebrafish serum. ELISA was used to quantify serum 11-KT levels across various genetic combinations of *amh*, *fshb*, and *lhb* in male zebrafish. The values are expressed as mean ± SEM (n ≥ 3), and different letters indicate statistical significance (P < 0.05). **(D)** Expression heatmap of androgen synthesis-related genes in zebrafish testis. RT-qPCR was conducted to quantify gene expression levels in testis, with *ef1a* serving as the normalization reference. The results are presented as fold changes compared to the control (*P < 0.05, **P < 0.01, ***P < 0.001). PG, primary growth; PV, pre-vitellogenic; LV, late vitellogenic.

## Co-expression evidence for Amh-Bmpr2a-Bmpr1bb pathway in zebrafish ovary

A fundamental requirement for establishing a functional signaling pathway is the spatiotemporal co-expression of its components. By combining analysis of isolated follicular layers with single-cell transcriptome data [48], we confirmed that *amh, bmpr2a, bmpr2b*, and *bmpr1bb* were predominantly expressed in the somatic follicle layer (granulosa and theca cells), whereas *bmpr1ba* was enriched in oocytes. Importantly, single-cell analysis revealed a specific subpopulation of granulosa cells co-expressing *bmpr2a* and *bmpr1bb*, thereby providing a foundation for their functional partnership. Although the co-expression was observed in a relatively small fraction of granulosa cells (~5.3%), this subpopulation likely represents a critical lineage of granulosa cells mediating Amh control of early folliculogenesis. The functional significance of this co-expression is supported by the strong genetic evidence, suggesting that the shared phenotypes observed in *bmpr2a* and *bmpr1bb* mutants may arise from dysfunction of this specific subpopulation of granulosa cells. Given the dynamic nature of folliculogenesis, it is plausible that the proportion of this granulosa cell subpopulation, characterized by *bmpr2a* and *bmpr1bb* co-expression, may change during follicle development.

Temporally, *amh* expression peaked during the PG-PV transition and declined thereafter, whereas *bmpr2a* and *bmpr2b* increased their expression progressively, reaching peak levels at FG stage, in agreement with our previous report [47]. The temporal expression mismatch between Amh and *bmpr2a* suggests that Bmpr2a is not dedicated solely to Amh. It is highly possible that Bmpr2a together with Bmpr2b also mediate signaling for other TGF-β superfamily ligands, especially other members of BMP family, during later stages of vitellogenic growth. This idea is supported by our previous study demonstrating that recombinant zebrafish Bmpr2a and Bmpr2b could both respond to recombinant zebrafish BMP4 [47].

## Genetic delineation of Amh-Bmpr2a-Bmpr1bb signaling pathway in zebrafish gonads

Unlike other vertebrates, including other fish species, zebrafish lacks a cognate type II receptor for Amh [35,38]. Our recent study revealed phenotypic similarities between *amh* and *bmpr2a* but not *bmpr2b* mutants [34,38], suggesting a potential role for Bmpr2a but not Bmpr2b as the Amh type II receptor in zebrafish. To further test this hypothesis, we examined the responsiveness of follicle cells from WT, *bmpr2a-/-*, and *bmpr2b-/-* females to recombinant zebrafish Amh. The results demonstrated clearly that while WT and *bmpr2b-/-* follicle cells exhibited significant Smad1/5/9 phosphorylation upon Amh stimulation, this response was completely absent in *bmpr2a-/-* cells. This strongly supports an essential role for Bmpr2a but not Bmpr2b in mediating Amh signaling in zebrafish.

While the type II receptor for Amh signaling has been studied in fish, less is known about the type I receptor. A previous study in zebrafish reported that a point mutation in *bmpr1bb* caused gonadal hypertrophy [39], a phenotype similar to that observed in *amh* and *bmpr2a* mutants. This finding suggests that Bmpr1bb may function as the type I receptor for Amh-Bmpr2a signaling. The zebrafish genome contains two paralogous copies of the Bmpr1b gene: *bmpr1ba* and *bmpr1bb*. To investigate whether Bmpr1ba also participates in Amh signaling and to further validate the role of Bmpr1bb, we generated mutant lines for both genes. The absence of any discernible phenotypes in *bmpr1ba-/-* fish ruled out its involvement in Amh signaling. In contrast, the loss of *bmpr1bb* resulted in gonadal hypertrophy, similar to but more severe than that observed in *amh* and *bmpr2a* mutants. The progressive phenotypic severity of the mutants along the Amh-Bmpr2a-Bmpr1bb pathway indicates that the receptors may also be involved in mediating signaling of other ligands in addition to Amh. Although we were unable to perform in vitro ligand–receptor activation assays for the type I receptors, primarily due to the limited availability of advanced follicles required for establishing follicle cell cultures, the in vivo genetic evidence is solid and compelling, supporting our proposal that Bmpr1bb, rather than Bmpr1ba, functions as the requisite type I receptor in this signaling pathway.

## Functional evidence for Amh-Bmpr2a-Bmpr1bb signaling in controlling zebrafish gonadal function and homeostasis

We previously demonstrated that mutation of *amh* gene in zebrafish caused severe gonadal hypertrophy in both males and females, characterized by the accumulation of early germ cells (increased proliferation/recruitment) and reduced meiotic

activities (decreased differentiation) [34]. Interestingly, simultaneous mutation of gonadotropin receptors (*fshr-/-* in females and *fshr-/-;lhcgr-/-* in males) completely rescued the phenotype of gonadal hypertrophy induced by *amh* mutation, indicating a critical role for gonadotropin signaling in this process [38]. Given that *bmpr2a* and *bmpr1bb* mutants phenocopied *amh* mutant, we hypothesized that disrupting gonadotropin signaling might also rescue the phenotypes observed in *bmpr2a* and *bmpr1bb* mutants. Consistent with this hypothesis, the absence of gonadotropin signaling also abolished gonadal hypertrophy in both *bmpr2a* and *bmpr1bb* mutants, mirroring the results observed in *amh* mutant [38]. These findings provide further functional evidence for an Amh-Bmpr2a-Bmpr1bb signaling pathway in zebrafish gonads and also indicate that gonadotropin signaling is tightly controlled by Amh through Bmpr2a/Bmpr1bb. If the Amh and gonadotropin pathways operated independently, one might expect an additive phenotype or partial rescue in the combined mutants. However, the complete abrogation of gonadal hypertrophy in the absence of gonadotropin receptors demonstrates that the hypertrophic phenotype is strictly dependent on gonadotropin signaling, supporting a model where Amh acts as a 'brake' on gonadotropin responsiveness (Fig 11).

To further dissect the interplay between Amh-Bmpr2a-Bmpr1bb signaling and gonadotropins, we next sought to determine which specific gonadotropin (FSH or LH) contributes to the observed phenotypes in *amh*, *bmpr2a*, and *bmpr1bb* mutants, *i.e.,* increased germ cell proliferation/recruitment (gonadal hypertrophy) and decreased differentiation (reduced meiotic activity and follicle activation). Our result demonstrated that gonadal hypertrophy occurred in *amh-/-* males and females in the presence of either *fshb* (FSH) or *lhb* (LH) alone, suggesting functional redundancy of FSH and LH in stimulating gonadal hypertrophy in the absence of Amh. This could be attributed to the ability of both zebrafish FSH and LH to activate the FSH receptor (Fshr) [52,53]. Interestingly, while gonadal hypertrophy was absent in triple mutant fish lacking both FSH and LH (*amh-/-;fshb-/-;lhb-/-*) at 90 dpf, the phenotype manifested at 240 dpf. This delayed onset of gonadal hypertrophy likely arises from the spontaneous receptor activity of Fshr and Lhcgr [45].

Interestingly, in contrast to our previous finding that the *fshb-/-;lhb-/-* double mutants displayed an all-male phenotype [54], we observed a small number of females in the *fshb-/-;lhb-/-* background with complete or partial loss of *amh* (-/- or +/-). This finding suggests that while eliminating both FSH and LH signaling primarily drives male development as

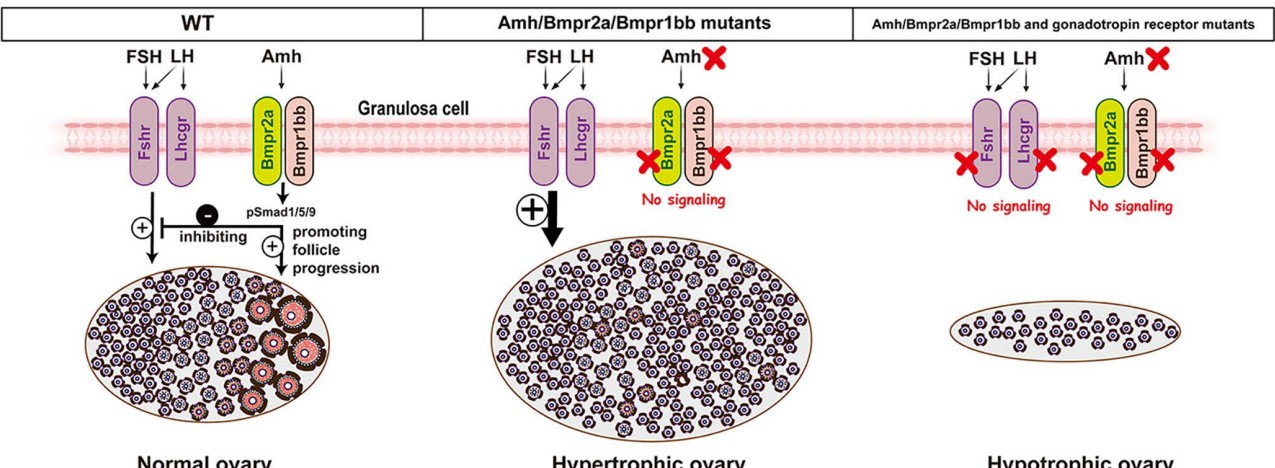

**Fig 11. Schematic model for the interplay between Amh-Bmpr2a-Bmpr1bb pathway and gonadotropin signaling in controlling gonadal homeostasis.** In wild-type zebrafish, Amh signals through Bmpr2a and Bmpr1bb to activate Smad1/5/9 in granulosa cells. This pathway serves dual functions: it acts as a brake on gonadotropin (FSH/LH) signaling to prevent excessive stimulation of the ovary; meanwhile, it promotes germ cell differentiation (meiosis in males and early follicle progression in females). Loss of any component of this pathway (*amh*, *bmpr2a*, or *bmpr1bb*) removes the restraint on gonadotropin signaling, resulting in hyperactive gonadotropin action and consequent gonadal hypertrophy. At the same time, early follicle development is impaired, particularly at the PG-PV transition. The hypertrophic phenotype is fully reversed by simultaneous loss of gonadotropin receptors (*fshr* and *lhcgr*), producing hypotrophic ovaries (testes in males), confirming that the overgrowth is driven by unchecked gonadotropin activity.

we proposed before [54], Amh may play a supporting role in the process of masculinization, and its partial or complete absence would permit a minority of individuals to remain as females during juvenile sex differentiation despite the lack of gonadotropin signaling. It should be noted that the follicles in these females were all arrested at PG stage, thereby phenocopying the FSH receptor (*fshr-/-*) mutant. However, their ovaries were considerably larger than those observed in the *fshr-/-* single mutant as well as the *bmpr2a-/-;fshr-/-* and *bmpr1bb-/-;fshr-/-* double mutants. These results provide additional evidence for the function of spontaneous Fshr activity in facilitating germ cell proliferation, as well as the role of ligand-stimulated Fshr activity in initiating follicle activation or the PG-PV transition.

In conclusion, this study provides comprehensive evidence that Bmpr2a and Bmpr1bb function as the type II and type I receptors, respectively, mediating Amh signaling in zebrafish gonads. Disruption of any component leads to unchecked gonadotropin signaling, resulting in gonadal hypertrophy. Under physiological conditions, gonadal homeostasis is maintained through a delicate balance between endocrine gonadotropins and local paracrine Amh signaling via Bmpr2a/Bmpr1bb (Fig 11).

## Materials and methods

### Ethics statement

All animal experiments were conducted according to the ethical standards and protocols approved by the Panel on Animal Research Ethics of University of Macau (AEC-13–002).

### Zebrafish husbandry and fish lines

The zebrafish (*Danio rerio*) were housed in the ZebTEC Multilinking Rack system (Tecniplast, Buguggiate, Italy). Wild-type (WT) AB strain and the following mutant lines were used in the study: *amh* (+62 bp; ZFIN line number: umo17) [34], *fshb* (-10 bp; umo1) [54], *lhb* (-5; umo2) [54], *fshr* (-11 bp; umo3) [52], *lhcgr* (-14 bp; umo4) [52]; *bmpr2a* (-5 bp; umo25) [38], and *bmpr2b* (+14 bp; umo26) [38].

The aquatic environment was maintained at $28 \pm 1°C$ with a pH of 7.5. A 14-h light period from 8:00 am to 10:00 pm was followed by a 10-h dark period. The zebrafish larvae were fed paramecia supplemented with chicken egg yolk for the first 5–10 days, transitioning to artemia from day 10–20 in nursery tanks in a room with an ambient temperature of $28 \pm 1°C$. Upon reaching an appropriate developmental stage, they were transferred to the main ZebTEC aquarium system. The fish were then fed twice daily via the Tritone Automatic Feeder (Tecniplast). Juvenile fish (less than one month old) received commercial fish feed B2, while adult fish (one month or older) were fed S1 (Marubeni Nisshin Feed, Tokyo, Japan), both supplemented with artemia.

### Preparation of recombinant zebrafish Amh

The recombinant zebrafish Amh protein was prepared from a stable CHO cell line established according to the protocol reported in our previous studies [47,53]. Briefly, the Amh-expressing CHO cells were seeded in 175-cm² flasks with 50 mL Ham's F-12K medium supplemented with 10% fetal bovine serum (FBS, Gibco) and cultured for three days at 37°C with 5% $CO_2$ (Gibco, Waltham, MA). Afterward, the cells were shifted to a serum-free medium and incubated for an additional five days at a reduced temperature of 28°C. Subsequently, the culture medium was collected and concentrated 200-fold using an Amicon Ultra 10000 MWCO filter unit (Merck Millipore, Burlington, MA). The same process was applied to the control cell line carrying only the empty vector.

### Primary culture of zebrafish follicle cells

Primary follicle cell culture was performed according to our previous studies [47,55]. Briefly, ten ovarian samples were obtained after 6 pm from female zebrafish aged over three months. The follicles were dispersed and washed a few times in L-15 medium (Gibco), followed by culturing at 28°C in 10-cm dishes containing M199 medium (Sigma-Aldrich, St. Louis, MO)

with 10% FBS. The media were changed after three days, and the culture continued until the sixth day. The proliferated follicle cells were then suspended by trypsinization and sub-cultured in a 24-well plate at a density of 1–2 × 10⁵ cells per well for 24 h. The cells were then subject to 24-h serum starvation in M199 medium before treatment with recombinant zebrafish Amh.

### Immunoblotting assay

Immunoblotting was performed as previously described [51]. Briefly, follicle cells from each well were collected using 45 µl SDS sample buffer (1x) for immunoblotting analysis. Following SDS-PAGE, proteins were transferred to a nitrocellulose membrane at 300 mA for 100 min. The membrane was blocked, washed, and incubated with primary and secondary antibodies, following standard immunoblotting protocols. Protein detection was performed using the ChemiDoc MP Imaging System (Bio-Rad). The primary antibodies used were pSmad1/5/9 (1:1000, #13820S, Cell Signaling Technology, Danvers, MA), actin (1:1000, #4967, CST), and GAPDH (1:1000, #2118S, CST). The secondary antibodies were HRP-linked anti-rabbit IgG (1:2000, #7074, CST) and HRP-linked anti-mouse IgG (1:2000, #7076S, CST).

### Establishment of mutant zebrafish lines for BMP type IB receptors (*bmpr1ba* and *bmpr1bb*)

The knockout of *bmpr1ba* and *bmpr1bb* genes in zebrafish was achieved using the CRISPR/Cas9 system as previously described [51]. The pDR274 plasmids (Addgene Plasmid #42250) carrying the necessary oligonucleotides were used to produce single guide RNAs (sgRNAs) with the MEGAscript T7 Transcription Kit (Life Technologies, Carlsbad, CA). A mixture containing targeting sgRNA (80 ng/µL) and Cas9 protein (600 ng/µL) (#M0646M; New England Biolabs, Ipswich, MA) was injected into zebrafish embryos at the one or two-cell stage using the Drummond Nanoject II device (Drummond Scientific, Broomall, PA). Mutagenesis was initially assessed at 24 hours post-fertilization (hpf) using high-resolution melting analysis (HRMA) and heteroduplex mobility assay (HMA). Subsequent genotyping of F0 and F1 adults was conducted on DNA extracted from caudal fins, with mutagenesis verified by sequencing. To generate homozygous F2 offspring for phenotypic evaluation, sibling F1 males and females harboring identical frameshift mutations were crossed with each other. The primers used for sgRNAs are listed in S1 Table.

### Genomic DNA extraction

The isolation of genomic DNA from single embryos, larvae or pieces of the caudal fin was performed using the NaOH extraction method [54,56]. This process involved addition of 40 µL NaOH (50 nmol/µL) to a sample tube containing an embryo or caudal fin tissue, which was then heated at 95°C for a duration of 10 min. The reaction was stopped by neutralization with 4 µL Tris-HCl (1 M, pH 8.0).

### Genotyping by high-resolution melt analysis (HRMA)

To genotype the mutants, HRMA, a real-time PCR-based technique, was employed using specific primers designed to amplify fragments encompassing the CRISPR target sites. The HRMA was performed on a CFX Real-Time PCR System (Bio-Rad, Hercules, CA), using the following protocol: an initial denaturation at 95°C for 3 min, followed by 40 cycles of 95°C for 15 sec, a pre-determined optimal annealing temperature of 60°C for 15 sec, and 72°C for 20 sec. A final melting curve analysis, ranging from 70°C to 95°C with increments of 0.2°C at each step, was then performed to distinguish wild-type and mutant alleles based on their distinct melting profiles. The data was analyzed using Precision Melt Analysis software (Bio-Rad). The primers used for HRMA are listed in S1 Table.

### Genotyping by heteroduplex motility assay (HMA)

To confirm the genotyping results obtained with HRMA, HMA was conducted using polyacrylamide gel electrophoresis (PAGE). PCR products from the HRMA analysis were analyzed by HMA. Briefly, 5 µL of the HRMA product was loaded

onto a 10% non-denaturing polyacrylamide gel and electrophoresed at 140 volts for 4 h. Following electrophoresis, the gel was stained with GelRed Nucleic Acid Gel Stain (#41003, Biotium, Hayward, CA) to visualize DNA bands. The gel was then imaged using the ChemiDoc Imaging System (Bio-Rad). The presence of heteroduplexes, formed by the annealing of wild-type and mutant strands, would manifest as additional bands with different electrophoretic mobility compared to the homoduplexes.

### Separation of oocyte and follicle layer

Oocyte and follicle layer separation was performed as previously described [41]. Briefly, full-grown (FG) follicles were carefully isolated in L-15 medium and then transferred to Cortland medium for a 30-min incubation at -20°C. This incubation step aimed to weaken the association between the oocyte and follicle layer (granulosa and theca cells). Following incubation, the follicle was held in place using a blunt-tipped forceps while the follicle layer was gently peeled away from the oocyte using a fine-tipped forceps, ensuring no damage to the oocyte. The isolated follicle layers and the corresponding oocytes from a batch of 10–20 follicles were pooled and immediately processed for RNA extraction using TRIzol (Invitrogen).

### RNA extraction and quantitative/semiquantitative RT-PCR

Total RNA was isolated from zebrafish ovarian follicles, follicle layers, denuded oocytes, and gonads using TRIzol (Invitrogen) according to the manufacturer's instructions. The concentration of RNA was determined by measuring the absorbance at 260 nm using a Nanodrop 2000 spectrophotometer (Thermo Scientific). For cDNA synthesis, total RNA (2 µg) was reverse transcribed using a cDNA synthesis kit (Invitrogen) in a 20 µL reaction volume. The reverse transcription reaction was performed at 37°C for 2 h according to the manufacturer's instructions.

Quantitative real-time RT-PCR (RT-qPCR) was performed to quantify gene expression levels. Specific primers were designed to amplify target genes and the housekeeping gene *ef1a* (see S1 Table for primer sequences). The qPCR reactions were set up in a total volume of 10 µL using cDNA, primers, and 2x SuperMix (Bio-Rad) on either a CFX384 or CFX96 Real-Time PCR Detection System (Bio-Rad). The amplification protocol consisted of an initial denaturation step at 95°C for 20 sec, followed by 40 cycles of denaturation at 95°C for 20 sec, annealing at 60°C for 20 sec, and extension at 72°C for 30 sec. Fluorescence signal detection was performed at 84°C for 8 sec after each cycle. Primer specificity was confirmed by melt curve analysis after amplification. Gene expression levels were quantified using the $2^{-\Delta\Delta CT}$ method with reference to the housekeeping gene *ef1a* and expressed as fold changes relative to the control group.

Semiquantitative RT-PCR was performed using a standardized reaction mix. Amplification was performed for 28–32 cycles at 95°C for 30 sec, 60°C for 30 sec, and 72°C for 40 sec. Optimal cycle numbers were determined following the method we previously reported [57]. PCR products were separated on an agarose gel, stained with GelRed Nucleic Acid Gel Stain (Biotium), and visualized on the ChemiDoc Imaging System (Bio-Rad). Primers for RT-qPCR and RT-PCR are listed in S1 Table.

### Histological analysis

Zebrafish from different genotypes were anaesthetized with MS-222 (Sigma-Aldrich). Photographs of each fish were taken using a Canon EOS 700D digital camera to document gross morphology. Standard body length (BL) was measured in centimeter using a ruler (photographed together with the fish), and body weight (BW) was measured in milligram using an analytical balance. The gonadosomatic index (GSI) was calculated as the ratio of gonad weight to body weight.

For histological examination, zebrafish tissues or bodies were fixed in Bouin's solution for one week at room temperature. The specimens were then dehydrated through a graded ethanol series and embedded in paraffin using an ASP6025S Automatic Vacuum Tissue Processor (Leica, Wetzlar, Germany). Tissue sections were cut to a thickness of

5 μm using a Leica microtome, deparaffinized, rehydrated, and stained with Hematoxylin and Eosin (H&E) for microscopic analysis. Finally, the tissue sections were observed under a Nikon ECLIPSE Ni-U microscope, and images were captured with a Digit Sight DS-Fi2 digital camera (Nikon, Tokyo, Japan).

### Follicle staging and quantification

To determine follicle composition in the ovary, we performed serial longitudinal sectioning of the entire fish at a thickness of 5 μm. From each fish, the three largest sections, spaced at least 60 μm apart, were chosen for follicle quantification using ImageJ. Only follicles with clearly visible germinal vesicle (GV, oocyte nucleus) were measured to ensure accurate diameter determination, as those without visible nuclei may represent tangential sections. Based on oocyte diameter and morphological characteristics such as the presence, size and distribution of cortical alveoli and yolk granules, we classified follicles into six stages: primary growth (PG or stage I, < 150 μm without cortical alveoli), previtellogenic (PV or stage II, ~ 250 μm with cortical alveoli), early vitellogenic (EV or early stage III, ~ 350 μm with yolk granules), mid-vitellogenic (MV or mid-stage III, ~ 450 μm), late vitellogenic (LV or late stage III, ~ 550 μm), and full-grown (FG, > 650 μm).

### Fertility assay

The fertility of the fish was assessed by the number of eggs spawned and the percentage of embryos that survived after natural mating with control fish. We tested at least 5 individuals for each genetic variant (+/+; +/-; -/-) at a 5-day interval. The ovulated eggs were counted within 3 h of spawning, and the survival of embryos was evaluated after 24 h. Fish that were unable to spawn or produce viable embryos after at least 10 attempts were considered infertile.

### Measurement of 11-KT levels in serum

To determine 11-ketotestosterone (KT) concentrations in serum, blood samples were collected from at least three fish per mutant or treatment group. Approximately 10–15 μL of blood were drawn directly from the heart of each fish using heparinized capillary tubes (KIMBLE #40C505, Capitol Scientific, Austin, TX), following methods described previously [43]. Blood samples were transferred to 1.5 mL tubes and allowed to clot at room temperature for 30 min to allow serum separation. Serum was then separated by centrifugation at 5000 rpm for 30 min at 4°C. The separated serum was carefully collected for measurement of serum 11-KT concentrations using a commercially available ELISA kit (#582751, Cayman Chemical Company, Ann Arbor, MI) according to the manufacturer's instructions.

### Analysis of *amh, bmpr2a/b* and *bmpr1ba/bb* expression at single cell level

To investigate the spatial expression patterns of *amh* and its potential type I (*bmpr1ba* and *bmpr1bb*) and type II receptors (*bmpr2a* and *bmpr2b*) in follicles, we reanalyzed a publicly available single-cell RNA sequencing (scRNA-Seq) dataset on early follicles from 40 days post-fertilization (dpf) zebrafish [48]. We downloaded gene expression data (plots and CSV files) from the Single Cell Portal (https://singlecell.broadinstitute.org/single_cell) and performed further analysis using an in-house R script based on Perl. We focused our analysis on types of follicle cells that express and co-express the type I and type II receptors (*bmpr2a/b* and *bmpr1ba/bb*). In addition, we also performed analysis focusing on expression and co-expression of the type II receptors (*bmpr2a* and *bmpr2b*) and gonadotropin receptors (*fshr* and *lhcgr*).

### Statistical analysis

Statistical analysis was performed using Prism software (GraphPad, San Diego, CA). Data are presented as means ± SEM. Statistical differences between groups were analyzed using Student's t test or one-way ANOVA, followed by multiple comparisons for pairwise comparisons. P-values less than 0.05 were considered statistically significant (*P < 0.05; **P < 0.01; ***P < 0.001; ns, not significant).

## Supporting information

**S1 Table. Primers used for CRISPR, HRMA and qPCR.**
(DOCX)

**S1 Fig. Phylogenetic analysis of Type I receptors for TGF-β superfamily.** The sequences were obtained from Gen-Bank and Ensemble databases. Sequence alignment and tree construction were performed by MEGA software using the Neighbor-Joining method. Zebrafish has two paralogous genes for both Bmpr1a (*bmpr1aa* and *bmpr1ab*) and Bmpr1b (*bmpr1ba* and *bmpr1bb*).
(TIF)

**S2 Fig. Establishment of *bmpr1ba* mutant in zebrafish.** (A) Genomic Structure of the zebrafish *bmpr1ba* Gene. The *bmpr1ba* comprises 14 exons (black boxes). The CRISPR/Cas9 target site, located on exon 6, is highlighted by a yellow box. We have established two mutant lines: one with a 5-bp deletion and the other with a 34-bp insertion. (B) The schematic representation of Bmpr1ba protein sequence structure. The mutations are predicted to result in premature stop codons. (C) HRMA assay for three genotypes. (D) HMA confirmation of different genotypes of *bmpr1ba* mutant.
(TIF)

**S3 Fig. Establishment of *bmpr1bb* mutant in zebrafish.** (A) Genomic Structure of the zebrafish *bmpr1bb* gene. The *bmpr1bb* gene consists of 15 exons (black boxes). Target sites for CRISPR/Cas9 editing are located in exon 1 and exon 11 (yellow boxes). Two mutant lines have been developed: one featuring a 20-bp deletion in exon 1 and another with an 11-bp deletion in exon 11. The deletions are indicated with red and green asterisks for the 20-bp and 11-bp deletions, respectively. (B) The schematic representation of Bmpr1bb protein sequence structure. The mutations result in premature stop codons. (C) HRMA assay for three genotypes. (D) HMA confirmation of different genotypes of *bmpr1bb* mutant.
(TIF)

**S4 Fig. Morphology and histology of *bmpr1ba* single mutants at 150 dpf.** (A) Morphology and histology of *bmpr1ba* (-5 bp) mutant. (B) Morphology and histology of *bmpr1ba* (+34 bp) mutant. Both males and females showed normal gonadal growth and gametogenesis. LV, late vitellogenic; FG, full-grown; SZ, spermatozoa.
(TIF)

**S5 Fig. Morphology and histology of *bmpr1bb* single mutants at 180 and 210 dpf.** (A) Morphology and histology of *bmpr1bb* (-11 bp) mutant at 180 dpf. (B) Morphology and histology of *bmpr1bb* (-20 bp) mutant at 210 dpf. EV, early vitellogenic; LV, late vitellogenic; SZ, spermatozoa.
(TIF)

**S6 Fig. Schematic illustration of differential signaling by different *bmpr1bb* mutants.** In WT, Amh signals through Bmpr2a and Bmpr1bb. The *bmpr1bb* (-11 bp) mutant retains the GS domain, allowing potential ligand binding. However, the mutant Bmpr1bb is unable to activate R-Smad and also inhibits other type I receptors, such as Bmpr1ba, from associating with Bmpr2a, thereby functioning as a dominant negative mutant. By comparison, the *bmpr1bb* (-20 bp) mutant lacks all functional domains, thereby allowing alternative type I receptors such as Bmpr1ba to couple with Bmpr2a in its absence.
(TIF)

**S7 Fig. Roles of gonadotropin signaling in the phenotypic expression of *bmpr1ba* and *bmpr1bb* mutations.** (A) Ovarian morphology and histology at 90 dpf in different genotypes. Double *bmpr1b* mutation (*bmpr1ba-/-;bmpr1bb-/-*) induced ovarian hypertrophy, characterized by the accumulation of early-stage follicles (PG and PV), while *fshr* mutation (*bmpr1ba+/-;bmpr1bb+/-;fshr-/-*) resulted in ovarian hypotrophy with only immature PG follicles. Triple mutation in both

*bmpr1b* and *fshr* (*bmpr1ba-/-;bmpr1bb-/-;fshr-/-*) rescued the hypertrophic phenotype observed in the double mutants (*bmpr1ba-/-;bmpr1bb-/-*). (B) Testis morphology and histology at 90 dpf in different genotypes. Double *bmpr1b* mutation (*bmpr1ba-/-;bmpr1bb-/-*) showed hypertrophy and disrupted spermatogenesis. Double mutation of *fshr* and *lhcgr* (*bmpr1ba+/-;bmpr1bb+/-;fshr-/-;lhcgr-/-*) resulted in testicular hypotrophy with minimal production of SZ. The quadruple mutants (*bmpr1ba-/-;bmpr1bb-/-;fshr-/-;lhcgr-/-*) showed no testicular hypertrophy displayed by the double *bmpr1b* mutant (*bmpr1ba-/-;bmpr1bb-/-*). One out of four individuals displayed PG follicles in the testis. PG, primary growth; PV, pre-vitellogenic; LV, late vitellogenic; SG, spermatogonia; SC, spermatocytes; SZ, spermatozoa.
(TIF)

## Acknowledgments

We extend our gratitude to Ms. Phoenix Un Ian LEI for her dedicated efforts in maintaining and managing the zebrafish facility. We acknowledge the Histology Core of the Faculty of Health Sciences for their invaluable technical support.

**Declaration of generative AI use in manuscript preparation**: In preparing this work, the author(s) used Copilot for language polishing to enhance clarity and readability in selected sentences. The use of AI was strictly limited to language refinement and did not involve data analysis, interpretation, referencing or drawing conclusions. All AI-generated suggestions were carefully reviewed and revised when necessary. The author(s) take full responsibility for the final content of this publication.

## Author contributions

**Conceptualization:** Yiming Yue, Wei Ge.

**Data curation:** Yiming Yue, Wei Ge.

**Formal analysis:** Yiming Yue, Wei Ge.

**Funding acquisition:** Xianqing Zhou, Wei Ge.

**Investigation:** Yiming Yue, Chu Zeng, Xin Zhang, Chao Bian, Kun Wu, Nana Ai, Wei Ge.

**Project administration:** Nana Ai, Wei Ge.

**Resources:** Zhiwei Zhang, Weiting Chen, Ling Lu, Wei Ge.

**Writing – original draft:** Yiming Yue, Wei Ge.

**Writing – review & editing:** Yiming Yue, Wei Ge.

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
