## [Decision Letter · Decision Letter 0]

10 Dec 2025

PGENETICS-D-25-01249

Genetic and Functional Characterization of AMH Signaling in Zebrafish - Evidence for Roles of Amh-Bmpr2a-Bmpr1bb Pathway in Controlling Gonadal Homeostasis

PLOS Genetics

Dear Dr. Ge,

Thank you for submitting your manuscript to PLOS Genetics. After careful consideration, we feel that it has merit but does not fully meet PLOS Genetics's publication criteria as it currently stands. Therefore, we invite you to submit a revised version of the manuscript that addresses the points raised during the review process.

We look forward to receiving your revised manuscript.

Kind regards,

Jeffrey J Essner

Academic Editor

PLOS Genetics

Pablo Wappner

Section Editor

PLOS Genetics

Aimée Dudley

Editor-in-Chief

PLOS Genetics

Anne Goriely

Editor-in-Chief

PLOS Genetics

**Journal Requirements:**

At this stage, the following Authors/Authors require contributions: Yiming Yue, Chu Zeng, Xin Zhang, Chao Bian, Zhiwei Zhang, Kun Wu, Weiting Chen, Xianqing Zhou, Ling Lu, Nana Ai, and Wei Ge. Please ensure that the full contributions of each author are acknowledged in the "Add/Edit/Remove Authors" section of our submission form.

The list of CRediT author contributions may be found here: https://journals.plos.org/plosgenetics/s/authorship#loc-author-contributions

https://journals.plos.org/plosgenetics/s/submission-guidelines#loc-parts-of-a-submission

5) We notice that your supplementary Figures, and Tables are included in the manuscript file. Please remove them and upload them with the file type 'Supporting Information'. Please ensure that each Supporting Information file has a legend listed in the manuscript after the references list.

Potential Copyright Issues:

- Please confirm (a) that you are the photographer of Figures 2, 3, 4, 6, 9, 10, S4, S5, and S7., or (b) provide written permission from the photographer to publish the photo(s) under our CC BY 4.0 license.

7) Please amend your detailed Financial Disclosure statement. This is published with the article. It must therefore be completed in full sentences and contain the exact wording you wish to be published.

**Reviewers' comments:**

Reviewer's Responses to Questions

**Comments to the Authors:**

Reviewer #1: It was known that zebrafish and other fish species lack Amhr2, the gene corresponding to Amh. Yue et al. investigated that bmp2a functions as the type 2 receptor and bmpr1bb as the type 1 receptor in AMH signaling. In addition to genetic analysis, they conducted detailed investigations, including demonstration that activation of SMAD in gonads by recombinant AMH requires Bmp2a.

It provides solid evidence demonstrating which receptors function as AMH signaling receptors in zebrafish lacking Amhr, particularly regarding the type 2 receptor.

Before publication, the reviewer requests some improvements, mainly additional explanation.

Major Comments

1. In Fig. 1E, the co-expression of type 1 receptors with bmp2a/2b is too low for 1ba and 1bb, making their relationship unclear. In Line 405, authors suggests bmpr1ba and 1bb are involved, but this figure lacks analysis. If extracting this data is feasible, please include it. If not, please explain the reason why bmpr1ba and 1bb were not examined.

2. AMH administration experiments were only performed on type 2 receptors, making type 1 receptor analysis relatively weak. Despite such smaller amount of evidence, the author concluded that bmpr1bb is the main type 1 receptor in AMH signaling with similar intensity as Bmp2a. Please explain why the authors can conclude Bmp1bb is the coupling type 1 receptor for Bmp2a in AMH signaling.

3. Line 572 states, “These results suggest that the Amh-Bmpr2a-Bmpr1bb pathway may work by suppressing gonadotropin signaling.” However, couldn't this be explained even if the Amh system and FSH/LH actions were independent?

Reviewer #2: In this manuscript, Ge and colleagues address an unresolved and important question regarding the hormone AMH in the zebrafish reproductive. Although zebrafish express Amh, they lack the canonical Amh receptor gene, leaving unclear how AMH signaling is mediated in this species. Through a series of genetic, biochemical, and gene expression analyses, the authors provide a compelling body of evidence that Amh acts through a non-canonical signaling pathway involving Bmp receptor 2a and 1bb. Furthermore, their data indicate that this Amh–Bmp2ra-1bb pathway requires Fshr- and Lhr-mediated signaling. Overall, the manuscript presents a robust amount of data, and the authors’ conclusions are convincing.

I have a few questions/suggestions that may further strengthen this strong and comprehensive study.

#1.

On page 14 (first paragraph), the authors report the co-expression of bmpr2a with bmpr2b, bmpr1ba, and bmpr1bb. According to the expression data provided, bmpr2a is co-expressed with bmpr2b in ~7% of granulosa cells, with bmpr1bb in ~4.3%, and with bmpr1ba in ~6.4%. The paragraph as written seems to imply that these co-expression patterns support the involvement of bmpr1ba and bmpr1bb, but not bmpr2b. However, the co-expression data alone do not exclude a possible role for bmpr2b. While the subsequent in vitro biochemical data do suggest that bmpr2b may indeed not be required, this specific paragraph should be reworded to avoid overinterpreting the expression patterns

# 2.

The description of the bmpr1bb -11 mutant on page 15 (last paragraph) and page 16 (first paragraph) raises an important interpretive issue. The -11 mutant fish have more severe phenotypes compared to -20. The authors hypothesize that the -11 allele produces a truncated receptor that may act in dominant-negative manner activity. A dominant-negative receptor may produce effects that differ from those of a simple loss-of-function mutation. This distinction is important because it can influence the interpretation of subsequent phenotypes and mechanistic conclusions. I suggest the authors more explicitly discuss how the dominant nature of the Δ11 allele may affect later analyses and whether some phenotypes might arise from non-physiological receptor activity.

#3. This is optional. The manuscript contains a remarkable and huge dataset, including multiple double and triple mutant combinations. While this is a major strength, it also makes the manuscript challenging to read. The MS is very long, and in some places- particularly in the Discussion section, there is redundancy with earlier sections. I recommend streamlining the manuscript by reducing repetitive descriptions and tightening the narrative. This would enhance readability for a broader audience while preserving the depth of the authors’ contributions.

If you want, I can also format this as a formal peer-review report letter or adapt the tone for a specific journal.

Reviewer #3: This is a comprehensive study investigating the signalling pathway involved in AMH-regulated gonadal development. Using a variety of techniques and extensive experimentation, the authors provide strong evidence to validate the previously proposed Amh-Bmpr2a-Bmpr1bb pathway and demonstrate its interaction with gonadotropins in controlling germ cell proliferation and differentiation. However, several issues need to be addressed:

1. Fig. 1C shows that amh is expressed at the highest level in PV and the level declines with follicle growth and reaches the lowest level in FG. However, both bmpr2a and bmpr1bb mRNA reach the highest level in FG. The discrepancy in expression patterns between amh and its receptors should be discussed. Are their other types for receptors for Amh? Do bmpr2a and bmpr1bb mediate the function of other TGF-beta family members?

2. Based on Fig. 1E (scRNA dataset, only a very small fraction (4.3%) of cells express bmpr2a+bmpr1bb, why do you think bmpr1bb KO has such dramatic effects?

3. In the Materials and Methods, the authors state that one-way ANOVA was used for data analyses. However, Fig. 2B and Fig 3B should be analyzed by two-way ANOVA, not one-way.

4. Fig. 5. For all western blots with pSmad1/5/9, total Smad1/5/9 must also be measured to confirm the activation, rather than a stimulation of Smad expression.

5. The results from the cross between bmpr mutants and fshr/lhcgr mutants are somewhat difficult to follow. Please include a diagram to show the different lines generated and key finding.

6. Include a summary diagram to illustrate how amh exerts its functions via bmprs and the interaction between this pathway and gonadotropin signaling pathways.

**Have all data underlying the figures and results presented in the manuscript been provided?**

Large-scale datasets should be made available via a public repository as described in the *PLOS Genetics*
data availability policy , and numerical data that underlies graphs or summary statistics should be provided in spreadsheet form as supporting information., and numerical data that underlies graphs or summary statistics should be provided in spreadsheet form as supporting information.

Reviewer #1: Yes

Reviewer #2: Yes

Reviewer #3: Yes

PLOS authors have the option to publish the peer review history of their article (what does this mean? ). If published, this will include your full peer review and any attached files.). If published, this will include your full peer review and any attached files.

**Do you want your identity to be public for this peer review?** For information about this choice, including consent withdrawal, please see our For information about this choice, including consent withdrawal, please see our Privacy Policy ..

Reviewer #1: No

Reviewer #2: No

Reviewer #3: No

**Figure resubmission:**
---

## [Decision Letter · Decision Letter 1]

4 Mar 2026

Dear Dr Ge,

We are pleased to inform you that your manuscript entitled "Genetic and Functional Characterization of AMH Signaling in Zebrafish - Evidence for Roles of Amh-Bmpr2a-Bmpr1bb Pathway in Controlling Gonadal Homeostasis" has been editorially accepted for publication in PLOS Genetics. Congratulations!

Yours sincerely,

Jeffrey J Essner

Academic Editor

PLOS Genetics

Pablo Wappner

Section Editor

PLOS Genetics

Aimée Dudley

Editor-in-Chief

PLOS Genetics

Anne Goriely

Editor-in-Chief

PLOS Genetics

BlueSky: @plos.bsky.social

Comments from the reviewers (if applicable):

Reviewer's Responses to Questions

**Comments to the Authors:**

Reviewer #1: All issues raised in the previous review have been satisfactorily resolved. The reviewer supports publication.

Reviewer #2: The authors have addressed all my comments.

Reviewer #3: The authors have addressed my concerns and the manuscript is now acceptable.

**Have all data underlying the figures and results presented in the manuscript been provided?**

Large-scale datasets should be made available via a public repository as described in the *PLOS Genetics*
data availability policy , and numerical data that underlies graphs or summary statistics should be provided in spreadsheet form as supporting information., and numerical data that underlies graphs or summary statistics should be provided in spreadsheet form as supporting information.

Reviewer #1: Yes

Reviewer #2: Yes

Reviewer #3: None

PLOS authors have the option to publish the peer review history of their article (what does this mean? ). If published, this will include your full peer review and any attached files.). If published, this will include your full peer review and any attached files.

**Do you want your identity to be public for this peer review?** For information about this choice, including consent withdrawal, please see our For information about this choice, including consent withdrawal, please see our Privacy Policy ..

Reviewer #1: No

Reviewer #2: No

Reviewer #3: No

**Data Deposition**

If you have submitted a Research Article or Front Matter that has associated data that are not suitable for deposition in a subject-specific public repository (such as GenBank or ArrayExpress), one way to make that data available is to deposit it in the Dryad Digital Repository . As you may recall, we ask all authors to agree to make data available; this is one way to achieve that. A full list of recommended repositories can be found on our . As you may recall, we ask all authors to agree to make data available; this is one way to achieve that. A full list of recommended repositories can be found on our website ..

http://datadryad.org/submit?journalID=pgenetics&manu=PGENETICS-D-25-01249R1

Additionally, please be aware that our data availability policy  requires that all numerical data underlying display items are included with the submission, and you will need to provide this before we can formally accept your manuscript, if not already present. requires that all numerical data underlying display items are included with the submission, and you will need to provide this before we can formally accept your manuscript, if not already present.

**Press Queries**

If you or your institution will be preparing press materials for this manuscript, or if you need to know your paper's publication date for media purposes, please inform the journal staff as soon as possible so that your submission can be scheduled accordingly. Your manuscript will remain under a strict press embargo until the publication date and time. This means an early version of your manuscript will not be published ahead of your final version. PLOS Genetics may also choose to issue a press release for your article. If there's anything the journal should know or you'd like more information, please get in touch via plosgenetics@plos.org ..

---

## [Editor Report · Acceptance letter]

PGENETICS-D-25-01249R1

Genetic and Functional Characterization of AMH Signaling in Zebrafish - Evidence for Roles of Amh-Bmpr2a-Bmpr1bb Pathway in Controlling Gonadal Homeostasis

Dear Dr Ge,

We are pleased to inform you that your manuscript entitled "Genetic and Functional Characterization of AMH Signaling in Zebrafish - Evidence for Roles of Amh-Bmpr2a-Bmpr1bb Pathway in Controlling Gonadal Homeostasis" has been formally accepted for publication in PLOS Genetics! Your manuscript is now with our production department and you will be notified of the publication date in due course.

With kind regards,

Livia Horvath

PLOS Genetics

On behalf of:
